# MotionMAR: Multi-scale Auto-Regressive Human Motion Reconstruction from Sparse Observations

Yuhua Luo [1 2 *]    Junsheng Zhang [1 2 *]    Mengyin Liu [1 2]    Xincheng Lin [1 2]    Ming Yan [1 2 3]    Zhudi Chen [1 2]
Chenglu Wen [1 2]    Lan Xu [4]    Siqi Shen [1 2]    Cheng Wang [1 2]

## Abstract

Human motion follows a temporal hierarchical structure, transitioning from low-frequency global trajectories to high-frequency details. Inspired by the success of multi-level autoregressive models in computer vision, we propose MotionMAR, a coarse-to-fine framework for motion reconstruction from sparse observations. It first estimates the global trajectory of human motion and then gradually refines the temporal details. This architecture consists of four integrated components. The Temporal Multi-scale Tokenization (TMT) VQ-VAE encodes the data at multiple temporal resolutions, separating semantic motion from minor jitters. The Motion Autoregressive Network (MAN) operates in this latent space, predicting motion across scales. It first establishes the global structure through coarse indices and then generates finer indices to recover specific details. Meanwhile, the Scale-Aware Control (SAC) module integrates sparse tracking data to ensure the generated output aligns with actual observations. The Motion Refinement Network (MRN) subsequently smooths consecutive poses and eliminates quantization artifacts. Experiments show that MotionMAR achieves state-of-the-art accuracy on the AMASS dataset, providing a reliable and structure-aware approach for motion reconstruction. The source code is publicly available at `http://www.lidarhumanmotion.net/motionmar/`.

## 1. Introduction

Human motion naturally follows a multi-level structure over time. Whether a casual walk or an expressive dance, a motion sequence is not just a flat series of joint positions. Rather, it is a multi-scale composition driven by intent, ranging from broad, low-frequency trajectories to quick, high-frequency details, which has also motivated hierarchical modeling in human pose and motion analysis (Li et al., 2021; Zheng et al., 2025). For instance, waving hello combines distinct time scales: a slow, broad motion guides the arm upward to reflect the main intent, while a faster, high-frequency motion controls the rapid shaking of the wrist. This coarse-to-fine structure means that broader trends naturally guide the finer details, ensuring the final motion is both meaningful and fluid.

However, traditional modeling methods, including many transformer-based approaches, often overlook this natural characteristic. They typically treat motion data as a flat sequence of frames or tokens, processing every time-step at a single scale (Lucas et al., 2022; Jiang et al., 2023; Zhang et al., 2023). Because they fail to separate the main motion envelope from the smaller, local variations, these single-scale strategies struggle to balance long-term stability with short-term precision. This limitation becomes especially obvious when dealing with sparse observations—such as inputs limited to VR headsets and controllers—where the model must reconstruct complex full-body dynamics from minimal data (Jiang et al., 2022; Zheng et al., 2023; Du et al., 2023; Castillo et al., 2023; Feng et al., 2024). Lacking a structural guide, these existing methods frequently suffer from floating artifacts, unwanted jitter, or ambiguous motion.

To bridge this gap, we introduce a new perspective by applying Visual Autoregressive (VAR) modeling—a generative framework known for its coarse-to-fine approach (Tian et al., 2024)—to align directly with the temporal nature of human motion. We observe a strong parallel between VAR's generative process and biological motion: just as VAR progressively adds image details to a broad outline, human motion evolves from a general intent to specific physical adjustments. Guided by this connection, we propose Mo-

---

*Equal contribution [1]Fujian Key Laboratory of Urban Intelligent Sensing and Computing, Xiamen University [2]Key Laboratory of Multimedia Trusted Perception and Efficient Computing, Ministry of Education of China, School of Informatics, Xiamen University [3]National Institute for Data Science in Health and Medicine, Xiamen University [4]ShanghaiTech University. Correspondence to: Siqi Shen <siqishen@xmu.edu.cn>.

*Proceedings of the 43rd International Conference on Machine Learning*, Seoul, South Korea. PMLR 306, 2026. Copyright 2026 by the author(s).

tionMAR, a multi-scale autoregressive framework designed to reconstruct high-quality human motion from sparse observations. The model operates through a next-scale generation strategy. It initially estimates coarse-scale indices to secure the overall trajectory, and then predicts finer-scale indices to fill in the detailed dynamics (Figure 1).

Simply transferring the original VAR framework—which was designed for static visual data—to human motion is not enough. To adapt this architecture for temporal dynamics, MotionMAR incorporates three key innovations. The foundation of the system is a TMT VQ-VAE, specifically tailored for motion sequences. Unlike the spatial scaling used in standard VAR, our model defines quantization scales based on time resolutions. This setup successfully separates broad semantic motion from minor jitters, maintaining strong temporal connections across different scales and reducing reconstruction errors. Our design is inspired by discrete latent modeling with VQ-VAE (Van Den Oord et al., 2017), which has also proved effective for motion representation learning (Lucas et al., 2022; Zhang et al., 2023). Building on this multi-scale foundation, the framework must also address the challenge of sparse inputs. To achieve this, we introduce a SAC Module. Rather than relying on static spatial structures, this module extracts continuous features from sparse trackers and aligns them to match the resolution of each generative scale. By embedding these aligned features into the prediction process, the module ensures that the generated motion remains strictly anchored to the observed control signals at both the broad trajectory and detailed dynamic scales. Once the autoregressive network generates these discrete indices, the system must translate them back into precise, continuous motion. To accomplish this, a MRN operates on the decoded pose space. Functioning as a temporal stabilizer, this module uses bidirectional recurrent connections to reduce quantization artifacts and smooth out local kinematic details, ensuring the resulting motion is physically fluid and accurate.

In summary, our main contributions are as follows:

1. We propose MotionMAR, a temporal coarse-to-fine autoregressive framework for sparse human motion reconstruction.

2. MotionMAR consists of three novel modules: a TMT VQ-VAE to decouple trajectories from jitter, a SAC Module to align tracking signals, and a MRN to smooth kinematics.

3. MotionMAR achieves state-of-the-art accuracy and temporal consistency on AMASS.

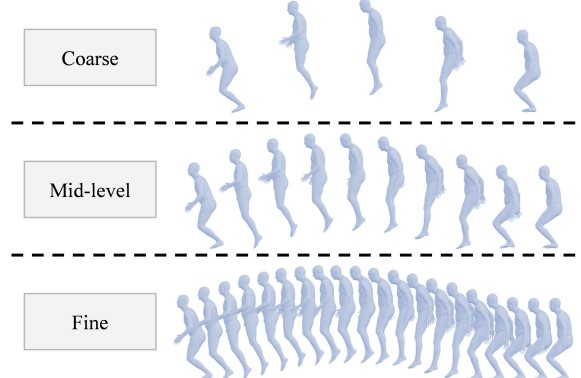

*Figure 1.* Visualization of MotionMAR's coarse-to-fine generation strategy. Rather than creating the motion in a single pass, the framework builds it sequentially across different time resolutions. The process starts by predicting a broad trajectory (Scale 1) to map out the general movement envelope. Following this initial outline, the model progressively adds mid-level dynamics (Scale 2) and high-frequency details (Scale 3) to complete the sequence.

## 2. Preliminaries and Related Work

### 2.1. Visual Autoregressive Modeling

Traditional autoregressive (AR) models (Esser et al., 2021; Yu et al., 2021; Lee et al., 2022; Van Den Oord et al., 2016; Van den Oord et al., 2016; Reed et al., 2017; Chen et al., 2018) for image generation operate by sequentially predicting the next token to construct the image patch by patch. The process begins with an encoder mapping the input image to a latent space, followed by a quantizer that discretizes the latent representation into a sequence of discrete tokens $(x_1, x_2, \ldots, x_T)$. The autoregressive model then predicts the subsequent token $x_t$ based on the preceding sequence $(x_1, x_2, \ldots, x_{t-1})$ and conditioning information $C$. The conditional probability can be expressed as:

$$p(x_1, x_2, \ldots, x_T \mid C) = \prod_{t=1}^{T} p(x_t \mid x_1, x_2, \ldots, x_{t-1}, C),$$

(1)

As highlighted in the Visual Autoregressive Modeling (VAR) framework (Tian et al., 2024; Ren et al., 2024b;a; Guo et al., 2025; Ma et al., 2024; Chen et al., 2024; Roheda et al., 2024; Yao et al., 2024; Li et al., 2024), the conventional "next-token prediction" approach (Yu et al., 2023; Nash et al., 2021; Ramesh et al., 2021) is fundamentally ill-suited for structured images, as it disrupts the inherent spatial structure and violates the autoregressive unidirectional dependency assumption. To address these limitations, VAR introduces an innovative "next-scale prediction" mechanism that encodes images into multi-resolution token maps and progressively generates content from low to high resolutions. By preserving 2D spatial correlations through multiscale representations and enabling parallel token generation

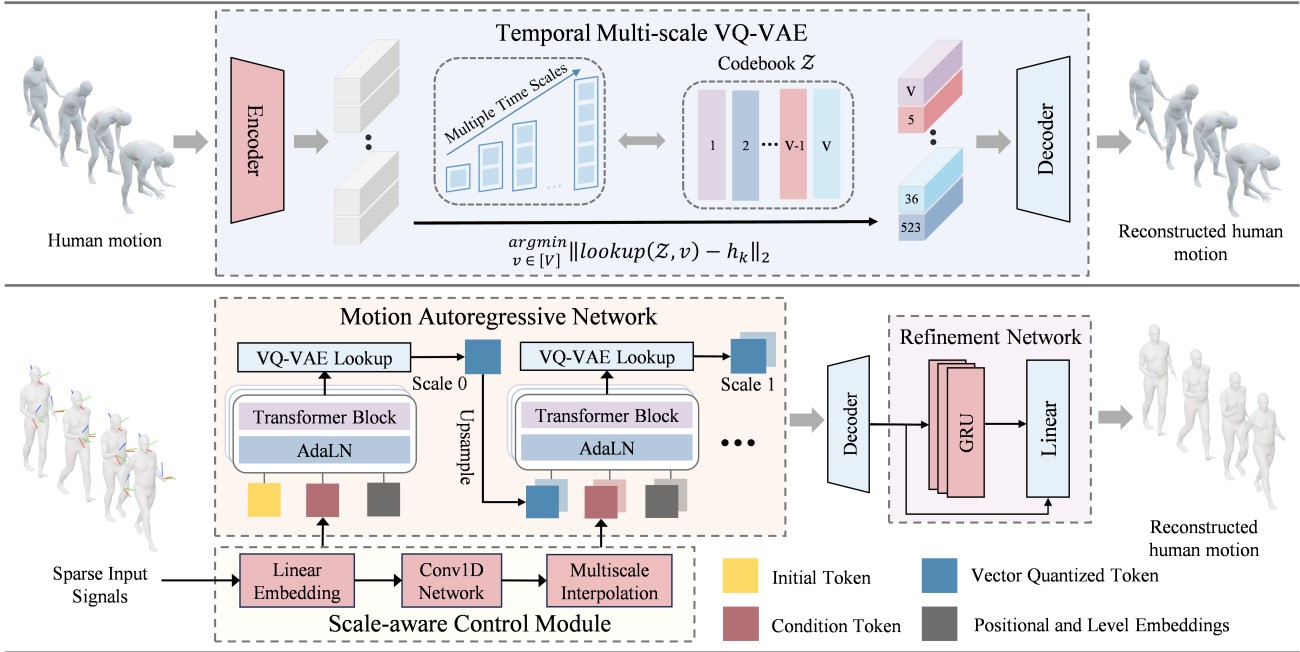

*Figure 2.* Overview of the Multi-scale Motion Autoregressive Network (MotionMAR). It consists of four core components: a Temporal Multi-scale VQ-VAE, a Scale-aware Control Module, a Motion Autoregressive Network, and a Refinement Network in the final stage.

within each resolution level, VAR achieves significant computational efficiency improvements, reducing complexity from $O(n^6)$ to $O(n^4)$.

## 2.2. Human Motion Reconstruction

Deep neural networks have achieved remarkable progress in 3D human motion reconstruction across diverse modalities (Chen et al., 2021; Saito et al., 2019; Wang et al., 2023; Ci et al., 2023; Zhu et al., 2023; Dou et al., 2023; Sun et al., 2023; Rempe et al., 2021; Kolotouros et al., 2019; Li et al., 2022; Yan et al., 2024; 2023; 2025; Wu et al., 2026; Li et al., 2026; Dai et al., 2023; 2024b). While foundational regression frameworks like HMR (Kanazawa et al., 2018) and VIBE (Kocabas et al., 2019) established strong baselines, they often struggle with complex motions and occlusions. To address these limitations, recent studies (Choi et al., 2020; Li et al., 2021; Zheng et al., 2025) have shifted towards structured modeling. For instance, AvatarPoser (Jiang et al., 2022) decouples global motion from local joint orientations, and SAGE (Feng et al., 2024) decomposes upper and lower limb motions via a progressive diffusion process for fine-grained reconstruction. RPM (Barquero et al., 2025) emphasizes robustness on practical VR motion data and provides an important benchmark for real-world sparse tracking scenarios. However, existing methods still lack explicit modeling of the intrinsic temporal hierarchy of human motion and sufficient multi-scale control, limiting their ability to fully capture subtle local dynamics and complex pose variations.

## 2.3. Generative and Quantization Models

With the successful application of generative models across various domains such as images, text, and speech (Chen et al., 2023; Ho et al., 2020; 2022; Huang et al., 2022; Yang et al., 2023), they have also been increasingly introduced into 3D human motion reconstruction tasks. BoDiffusion (Castillo et al., 2023) generates balanced full-body motion sequences through dual conditioning on temporal and spatial dimensions. Meanwhile, methods such as PoseGPT (Lucas et al., 2022), MotionGPT (Jiang et al., 2023), and T2M-GPT (Zhang et al., 2023) leverage VQ-VAE (Van Den Oord et al., 2017) to encode human motion into a discrete latent space and perform autoregressive prediction of latent tokens within this space, enabling more structured and coherent motion generation. As far as we know, MotionMAR is the first approach adapting a multi-scale autoregressive method for motion reconstruction from sparse observations.

## 3. Methodology

### 3.1. Problem Formulation

Our goal is to predict high-fidelity full-body motion given sparse tracking signals derived from standard commercial VR/AR devices. Specifically, we utilize three sensors mounted on the head, left hand, and right hand to capture the corresponding 6-DoF rigid body transformations (including 3D rotation and translation in the global coordinate system).

**Sparse Inputs.** We represent the raw tracking signals as a time-dependent sequence $X_{raw}(t) = (X_h(t), X_l(t), X_r(t))$, corresponding to the head ($h$), left hand ($l$), and right hand ($r$). To help the model learn motion dynamics more effectively, we follow standard practices (Feng et al., 2024; Jiang et al., 2022) and expand these initial inputs into an augmented observation matrix $X \in \mathbb{R}^{T \times 3 \times 18}$. Within this structured matrix, each of the three sensors is described by an 18-dimensional feature vector. This vector combines the 6-DoF pose parameters with their corresponding linear and angular velocities, equipping the network with a clearer dynamic context for accurate reconstruction.

**Motion Representation.** For the target full-body motion, we adopt the 6D rotation representation for joint orientations. As demonstrated in prior studies (Zhou et al., 2019; Feng et al., 2024), continuous 6D representations offer better mathematical continuity for neural network training compared to 3D Euler angles or quaternions. Consequently, we represent the full-body motion sequence as $\theta \in \mathbb{R}^{T \times 22 \times 6}$, covering the rotations of 22 key joints over T frames.

### 3.2. Methodology Overview

To address the highly ambiguous nature of sparse motion reconstruction, we introduce MotionMAR, a multi-scale autoregressive framework. As illustrated in Figure 2, the system adopts a temporal coarse-to-fine approach. It begins by estimating a broad global trajectory, then progressively refines the local temporal details to achieve higher accuracy. This progressive generation is driven by three interconnected components: a Temporal Multi-scale VQ-VAE (Section 3.2.1), a Motion Autoregressive Network (MAN) (Section 3.2.2), and a Motion Refinement Network (Section 3.2.3).

#### 3.2.1. TEMPORAL MULTI-SCALE VQ-VAE

This section details how we map continuous human motion into a multi-scale discrete latent space. Instead of encoding the motion sequence at a single time resolution, we introduce a temporal coarse-to-fine decomposition. By structuring the data in this manner, the framework separates the motion into a feature pyramid, descending from broad semantic trends to detailed local kinematics. This multi-scale organization ultimately provides the structural foundation required for our autoregressive generation process.

As illustrated in Figure 2, the Multi-scale VQ-VAE is built around a Transformer-based encoder (E) and decoder (D), operating alongside a multi-scale quantizer. The data flow begins with the encoder, which processes a human motion sequence $\theta = \{\theta_i\}_{i=1}^{T}$ and compresses it into a continuous latent representation $H \in \mathbb{R}^{t \times d}$. In this space, $t$ represents the temporally downsampled length and $d$ indicates the

feature dimension. Following this compression, the multi-scale quantizer $\{Q_k\}_{k=1}^{K}$ maps these continuous features into discrete tokens $q \in [V]^{t \times d}$.

**Temporal Multi-scale Tokenization.** While conventional autoregressive models for images focus on spatial scales, our approach adapts to the continuous flow of human motion by building a hierarchy based on time resolutions. The encoding process structures the motion across multiple scales, beginning with coarse, low-frequency representations that map out the global trajectory and gradually expanding into finer, high-frequency details. Fast, localized motion are directly guided by the stable foundation established in the coarser stages (Figure 2). Aligning with this structure, our network predicts data one time scale at a time, steadily refining the sequence from a broad motion envelope to precise physical actions.

To map these continuous features to discrete symbols, we employ an iterative residual quantization process across $K = 3$ temporal scales ($t_k \in \{T/4, T/2, T\}$) using a single, shared learnable codebook $\mathcal{Z} \in \mathbb{R}^{V \times d}$, where $V = 1024$ and $d = 64$. Given an input motion latent feature of length $T$, the residual feature is initially set to this input. At each scale $k$, the current residual is downsampled to the target resolution $t_k$ via 1D linear interpolation, producing the continuous feature $h_k$. Both $h_k$ and the codebook embeddings are then $L_2$-normalized, enabling the retrieval of the nearest codebook vectors, denoted as $q_k$, based on cosine similarity. These quantized vectors $q_k$ are interpolated back to the original length $T$ and passed through a 1D residual convolution block ($\Phi$) to smooth temporal discontinuities. This smoothed feature is added to the overall reconstruction and subtracted from the current residual, generating the updated input for the subsequent, finer scale.

$$q_k = \left( \underset{v \in [V]}{\arg\min} \|\text{lookup}(\mathcal{Z}, v) - h_k\|_2 \right) \in [V], \quad (2)$$

Here, $k$ indexes the temporal scale, and $\text{lookup}(\mathcal{Z}, v)$ returns the $v$-th vector in the codebook $\mathcal{Z}$.

Next, we will introduce more details about the Temporal Multi-Scale VQ-VAE. We use the index $q_k$ to retrieve the discrete latent vector $z_k = \text{lookup}(\mathcal{Z}, q_k)$. The decoder then takes $z_k$ together with the processed sparse observations to reconstruct the human motion pose. Finally, the encoder, decoder, and hierarchical quantizer are optimized jointly with the following losses.

$$\mathcal{L}_{all} = \lambda_1 \mathcal{L}_{rec} + \lambda_2 \mathcal{L}_{vq} + \lambda_3 \mathcal{L}_{loc} + \lambda_4 \mathcal{L}_h, \quad (3)$$

$$\mathcal{L}_{rec} = Smooth_{\mathcal{L}_1}(\theta, \hat{\theta}), \quad (4)$$

$$\mathcal{L}_{vq} = \sum_{k=1}^{K} \| sg[z_k] - h_k \|_2 + \beta \| z_k - sg[h_k] \|_2, \quad (5)$$

$$\mathcal{L}_{loc} = Smooth_{\mathcal{L}_1}(J, \hat{J}), \quad (6)$$

$$\mathcal{L}_{h} = Smooth_{\mathcal{L}_1}(\theta_{hand}^{glo}, \hat{\theta}_{hand}^{glo}), \quad (7)$$

where $\mathcal{L}_{rec}$ denotes the reconstruction loss of the SMPL pose parameters ($\theta$); $sg$ denotes stop gradient; $\mathcal{L}_{vq}$ represents the total loss accumulated across all hierarchical quantization levels during the vector quantization process; $\beta$ is a hyperparameter; $\mathcal{L}_{loc}$ refers to the joint loss of SMPL in the local coordinate system; $J$ and $\hat{J}$ individually represent the joint points of each sequence in the SMPL model; and since sparse observations provide hand data in the global coordinate system, we use $\mathcal{L}_h$ to calculate the hand pose loss in the global coordinate system, aiming to precisely align the global hand position. In addition, $\lambda_i, i \in [1, 4]$ represents the weight parameters for each loss.

### 3.2.2. MOTION AUTOREGRESSIVE NETWORK

Once the multi-scale VQ-VAE maps the continuous motion $\theta$ into a discrete temporal pyramid, the Motion Autoregressive Network (MAN) takes over as the generative core. Conditioned solely on the augmented sparse observations $X$, MAN reconstructs the full motion sequence by progressively sampling indices from this latent space. Rather than following the traditional spatial SMPL (Loper et al., 2023) kinematic tree, we structure the motion generation across temporal layers, moving from low-frequency global trends down to high-frequency local details. This hierarchical approach ensures that broad global structures strictly guide the generation of fine-grained poses, leading to physically stable and accurate results.

Built on a GPT-2-style (Radford et al., 2019) transformer architecture, the network begins by encoding the sparse observations into initial conditioning embeddings. The generation process then operates recursively across scales, as illustrated in Figure 2. At each stage, the quantized features from the current scale are upsampled to match the temporal resolution of the next scale. The network processes these upsampled features to predict the token indices for that finer scale. To prevent early prediction errors from cascading through the hierarchy during training, we apply teacher forcing. Supplying the network with ground-truth outputs instead of its own previous predictions allows it to learn more reliable mappings across layers without accumulating errors. To optimize the network, we apply a cross-entropy loss between the predicted indices $\hat{q}_k$ and the ground-truth

indices $q_k^{gt}$ at each scale, which are extracted by passing real human motion through the VQ-VAE. The final objective sums this loss across all $K$ scales:

$$\mathcal{L}_{MAN} = \sum_{k=1}^{K} \mathrm{CE}(\hat{q}_k, q_k^{gt}). \quad (8)$$

**Scale-aware Control Module.** Standard VAR frameworks apply identical conditioning inputs across all hierarchical scales, limiting their ability to adapt to varying temporal granularities. To overcome this, our Scale-aware Control Module dynamically adjusts the network's conditioning at each scale to match the specific temporal requirements of the motion.

The process begins by projecting flattened sparse tracking signals (such as joint positions from Head-Mounted Display) into a high-dimensional latent space, where a 1D convolutional block extracts continuous local temporal features. Since the autoregressive network operates on discrete, multi-resolution tokens, these continuous signals require temporal alignment. We achieve this through linear interpolation, resampling the extracted features to match the exact temporal resolution ($t_k$) of each scale defined by the Temporal VQ-VAE. This alignment yields a pyramid of control features $\{C_0, C_1, \ldots, C_K\}$, perfectly synchronizing the conditioning signal with the token sequence length at every scale.

These aligned features are then integrated into the Motion Autoregressive Network through two complementary pathways. For local scale-wise guidance, the scale-specific feature $C_k$ is added element-wise to the input token embeddings at each autoregressive step $k$. This frame-aligned constraint allows coarse-scale features to establish the global trajectory while fine-scale features drive local dynamic refinement. Concurrently, a global context modulation maintains overall consistency in motion style and semantics. By applying global average pooling to the encoded features, we generate a static context vector that modulates the Adaptive Layer Normalization (AdaLN) parameters across all transformer blocks throughout the generation process.

### 3.2.3. MOTION REFINEMENT NETWORK

While the Motion Autoregressive Network predicts discrete indices, our ultimate goal is to recover continuous motion parameters. Decoding these indices back into poses via the VQ-VAE often introduces quantization artifacts and temporal jitter. To resolve this, we apply a Motion Refinement Network as a post-processing module to optimize the initial decoded sequence (Figure 2).

Operating directly on the continuous pose sequence, this refinement module is built as a multi-layer Bidirectional Gated Recurrent Unit (Bi-GRU). By aggregating temporal context

from both past and future frames simultaneously, the bidirectional architecture ensures globally coherent smoothing and corrects geometric inconsistencies.

Formally, the network applies a residual learning strategy. It takes the initial decoded motion $\hat{\theta} \in \mathbb{R}^{T \times D_{\text{SMPL}}}$ (where $D_{\text{SMPL}} = 132$ represents the dimension of SMPL parameters) and predicts a corrective offset $\Delta\theta = \text{Bi-GRU}(\hat{\theta})$. This residual dynamically adjusts the poses to satisfy physical constraints. The final refined motion is then computed by adding this corrective term back to the input: $\hat{\theta}_{final} = \hat{\theta} + \Delta\theta$.

## 4. Experiments

### 4.1. Datasets and Metrics

To evaluate MotionMAR, we test its ability to reconstruct full-body motion from sparse tracking signals. In line with standard experimental setups (Rempe et al., 2021; Aliakbarian et al., 2022; Jiang et al., 2022; Zheng et al., 2023), we limit the input to head and hand sensor data to simulate a typical VR/XR environment. Because this task requires recovering high-fidelity sequences from minimal kinematic constraints, it rigorously challenges the model's capacity to infer missing information.

For this evaluation, we use the AMASS dataset (Mahmood et al., 2019), a large-scale archive that standardizes diverse motion capture collections into a common SMPL format. Following the protocols defined in (Feng et al., 2024; Du et al., 2023), we assess our approach across three distinct settings derived from AMASS subsets:

In the first setting ($S1$), we combine the CMU (University), BMLrub (Troje, 2010), and HDM05 (Müller et al., 2007) datasets, randomly splitting them into 90% for training and 10% for testing. The input for this setting consists of sparse observations from the head and both hands. In the second setting ($S2$), we utilize the same dataset split as the first but augment the input with root joint observations, resulting in a 4-tracker setup. In the third setting ($S3$), we introduce a large-scale evaluation scenario to assess model generalization. We compile a comprehensive dataset comprising CMU (University), MPI Limits (Akhter & Black, 2015), TotalCapture (Trumble et al., 2017), Eyes Japan (Ltd.), KIT (Mandery et al., 2015), BioMotionLab (Troje, 2010), BMLmovi (Ghorbani et al., 2020), EKUT (Mandery et al., 2015), ACCAD (Osu accad), MPI Mosh (Loper et al., 2014), SFU (Sfu motion capture database), and HDM05 (Müller et al., 2007). Adopting the same random partition strategy as ($S1$), we split this aggregated dataset into 90% for training and the remaining 10% for testing. The input configuration for this setting remains consistent with the first setting.

To evaluate model performance, we adopt metrics from (Feng et al., 2024) covering three dimensions: accuracy, consistency, and smoothness. We measure joint-wise rotation and position accuracy via MPJRE [$degrees$] (mean per joint rotation error) and MPJPE [$cm$] (mean per joint position error), with specific focus on Upper PE (upper MPJPE), Lower PE (lower MPJPE), Root PE (root MPJPE) and Hand PE (hand MPJPE). To assess motion quality, we employ MPJVE [$cm/s$] (mean per joint velocity error) to quantify velocity deviation and Jitter [$10^2 \ m/s^3$] to evaluate temporal smoothness through acceleration variations. For all metrics, lower values signify superior performance.

### 4.2. Quantitative and Qualitative Results

To validate the effectiveness of our approach, we benchmark it against representative methods from various paradigms. These include the traditional optimization-based Final IK (RootMotion, 2018), the matching-based CoolMoves (Ahuja et al., 2021), the MLP-based LoBSTr (Yang et al., 2021), and the VAE-based VAE-HMD (Dittadi et al., 2021). We also compare against Transformer-based regression approaches, specifically AvatarPoser (Jiang et al., 2022) and AvatarJLM (Zheng et al., 2023). Furthermore, we evaluate our model against recent generative frameworks, including Diffusion-based methods (AGRol (Du et al., 2023), SAGE (Feng et al., 2024)) and Autoregressive models (MAGE (Lin et al., 2026), HiPART (Zheng et al., 2025), RPM (Barquero et al., 2025)). It is worth noting that MAGE and HiPART were originally designed for Reinforcement Learning and Human Pose Estimation, respectively. To enable a fair comparison, we adapted these methods to the task of human motion reconstruction.

As shown in Table 1, our method outperforms existing approaches on most evaluation metrics, demonstrating the effectiveness of our MotionMAR framework. First, traditional optimization-based methods like Final IK and MLP-based approaches like LoBSTr exhibit a significant performance gap compared to the other four categories. While the VAE-based method, VAE-HMD, shows clear improvements over the former two, it still suffers from notable deficiencies in MPJRE and MPJPE. We argue that relying solely on VAE architectures is insufficient for modeling complex human motion. Among Transformer-based regression methods, AvatarPoser yields suboptimal results in MPJRE and MPJPE. In contrast, AvatarJLM optimizes joints at different hierarchical levels, achieving strong performance in MPJPE, MPJVE, Hand PE, and Root PE; however, it exhibits relatively high error in MPJRE. We attribute this to inadequate optimization of the root node during the Transformer-based modeling phase.

Regarding diffusion-based methods, although AGRoL achieves a low MPJVE score in the offline setting, it processes the entire sparse observation sequence glob-

*Table 1.* Results for Human Motion Reconstruction under setting S1.

| Method | MPJRE ↓ | MPJPE ↓ | MPJVE ↓ | Hand PE ↓ | Upper PE ↓ | Lower PE ↓ | Root PE ↓ | Jitter ↓ |
|---|---|---|---|---|---|---|---|---|
| Final IK (RootMotion, 2018) | 16.77 | 18.09 | 59.24 | - | - | - | - | - |
| LoBSTr (Yang et al., 2021) | 10.69 | 9.02 | 44.97 | - | - | - | - | - |
| VAE-HMD (Dittadi et al., 2021) | 4.11 | 6.83 | 37.99 | - | - | - | - | - |
| AvatarPoser (Jiang et al., 2022) | 3.08 | 4.18 | 27.70 | 2.12 | 1.81 | 7.59 | 3.34 | 14.49 |
| AvatarJLM (Zheng et al., 2023) | 2.90 | 3.35 | 20.79 | 1.24 | 1.42 | 6.14 | 2.94 | 8.39 |
| AGRoL (Online) (Du et al., 2023) | 2.96 | 4.26 | 79.12 | 1.51 | 1.73 | 7.91 | 3.78 | 84.79 |
| AGRoL (Offline) (Du et al., 2023) | 2.66 | 3.71 | 18.59 | 1.31 | 1.55 | 6.84 | 3.36 | 7.26 |
| SAGE (Feng et al., 2024) | 2.53 | 3.28 | 20.62 | 1.18 | 1.39 | 6.01 | 2.95 | 6.55 |
| MAGE (Lin et al., 2026) | 2.89 | 3.83 | 64.91 | 1.40 | 1.69 | 8.94 | 3.36 | 53.42 |
| HiPART (Zheng et al., 2025) | 2.75 | 3.71 | 98.19 | 1.74 | 1.93 | 8.54 | 3.16 | 107.1 |
| RPM (Barquero et al., 2025) | 3.25 | 4.08 | 19.29 | 3.61 | 2.17 | 6.83 | 3.47 | **4.20** |
| **MotionMAR (Ours)** | **2.39** | **2.82** | **16.23** | **0.83** | **1.22** | **5.13** | **2.58** | 5.17 |

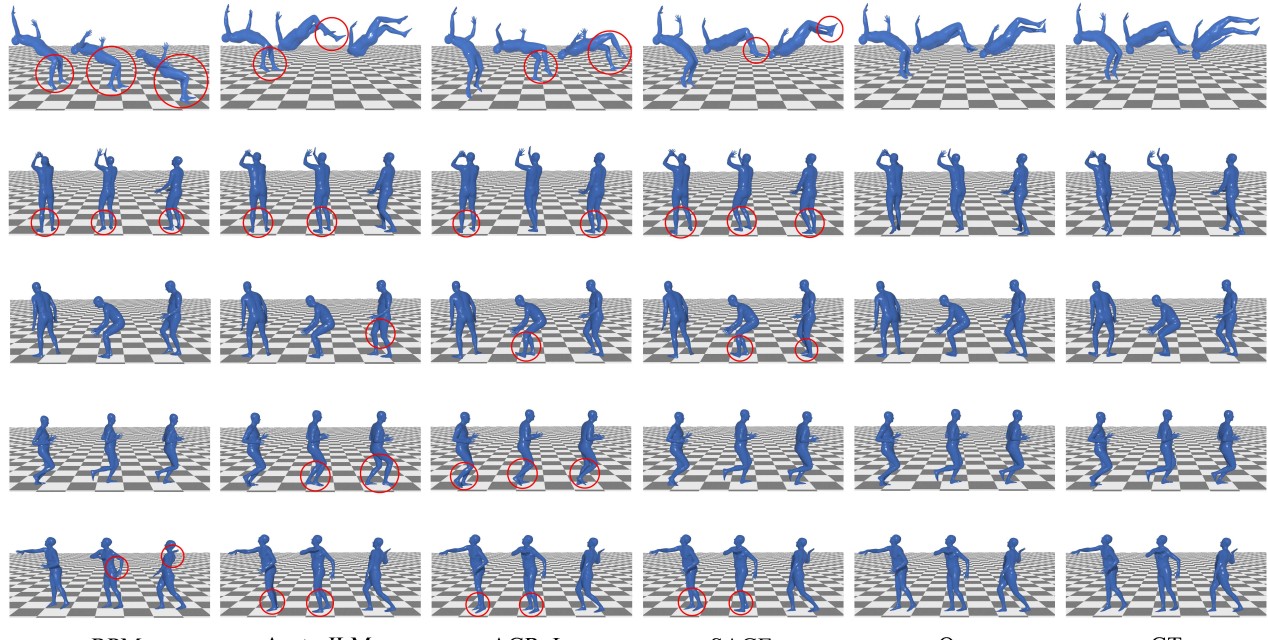

*Figure 3.* Visual comparison of MotionMAR against baseline methods for Human Motion Reconstruction under setting $S1$.

ally—leveraging both past and future temporal information—as noted in (Feng et al., 2024). However, its performance degrades significantly in the online setting, likely due to the constraints of real-time application scenarios. SAGE decouples human motion into upper and lower body components, progressively generating their latent representations via a diffusion process. These representations are then fused and decoded by a full-body decoder. Despite its strong performance in motion reconstruction tasks, SAGE still demonstrates a lack of precision in lower-body regions as shown in the second and third rows of Figure 3.

Finally, regarding autoregressive approaches, while MAGE and HiPART were not originally designed for human motion reconstruction, they show promising results in pose estimation. HiPART models human joints spatially on a single-frame basis, neglecting holistic temporal motion, which results in substantial jitter. Conversely, MAGE models trajectories, thereby ensuring overall consistency. RPM achieves the best results in Jitter scores, demonstrating the advantage of Rolling Prediction for optimizing temporal motion; however, it overlooks the optimization of the human structural pose itself, leading to suboptimal performance in reconstruction metrics, the visual comparison in Figure 3

*Table 2.* Results for Human Motion Reconstruction under setting S2.

| Method | MPJRE ↓ | MPJPE ↓ | MPJVE ↓ |
|---|---|---|---|
| Final IK | 12.39 | 9.54 | 36.73 |
| CoolMoves | 4.58 | 5.55 | 65.28 |
| LoBSTr | 8.09 | 5.56 | 30.12 |
| VAE-HMD | 3.12 | 3.51 | 28.23 |
| AvatarPoser | 2.59 | 2.61 | 22.16 |
| AvatarJLM | 2.40 | 2.09 | 17.82 |
| AGRoL | 2.25 | 2.17 | 16.26 |
| SAGE | 2.10 | 1.88 | 14.79 |
| RPM | 2.53 | 2.19 | 17.34 |
| **MotionMAR (Ours)** | **1.98** | **1.71** | **13.20** |

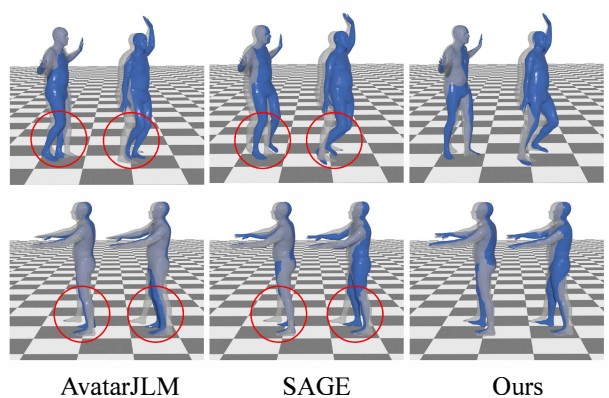

AvatarJLM      SAGE      Ours

*Figure 4.* Visualization results on real data. Blue indicates the predicted results, while white represents the ground truth.

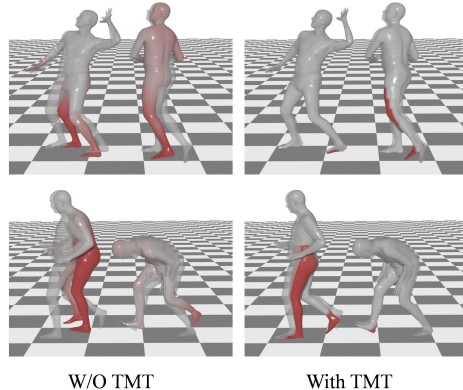

W/O TMT      With TMT

*Figure 5.* Visualization comparison for the Temporal Multi-scale Tokenization. The darker the red color, the greater the deviation between the predicted result and the ground truth.

also confirms this finding. In contrast, our method, MotionMAR, achieves highly competitive results in capturing both structural human pose and coherent full-body motion. No-

tably, we observe satisfactory improvements in Hand PE and Lower PE. The results from the comparative experiments show that our method achieves a high degree of consistency with the ground truth (GT).

Additionally, to verify the generalization capability of MotionMAR, we conducted quantitative comparisons under two alternative settings, S2 and S3. The results, presented in Table 2 and Table 3, demonstrate that our approach consistently retains its competitive edge and superior reconstruction accuracy across diverse experimental configurations.

We also evaluated our model on real-world data. To ensure a fair comparison, we directly employed the real-world dataset released by (Zheng et al., 2023). As shown in Figure 4, our method exhibits superior alignment with the Ground Truth, particularly in the hand and leg regions. This qualitative comparison demonstrates that our approach achieves better reconstruction performance in real-world scenarios.

### 4.3. Ablation Study

To validate the contribution of each core component to human motion reconstruction performance in our method, we conducted a systematic ablation study. Table 4 presents the evaluation results on the AMASS dataset under setting S1 for different combinations of key modules, including Motion Refinement Network (MRN), Scale-aware Control Module (SAC), and Temporal Multi-scale Tokenization (TMT).

The experimental results demonstrate the distinct contributions of each component. First, the Motion Refinement Network (MRN) consistently improves fluidity metrics, specifically MPJVE and Jitter. This indicates that MRN effectively mitigates quantization artifacts, thereby enhancing the realism of the generated motions. Furthermore, adding the Scale-aware Control Module (SAC) yields substantial gains in Root and Hand PE. This improvement stems from the module's temporal alignment strategy, which ensures predicted motions remain strictly anchored to sparse control signals at each autoregressive scale.

Most significantly, the Temporal Multi-scale Tokenization (TMT) provides the largest performance boost across MPJPE, Upper PE, and Lower PE. By modeling motion at varying resolutions, it captures the essential coarse-to-fine kinematic structure, maintaining lower reconstruction errors. Additionally, Figure 5 presents a visual comparison of our Temporal Multi-scale Tokenization, demonstrating that it significantly improves the results of human motion reconstruction. In summary, the complete MotionMAR framework successfully integrates these complementary strengths: TMT lays the hierarchical foundation, SAC ensures alignment with observations, and MRN acts as the final stabilizer. The superior performance of the full model confirms that

*Table 3.* Results for Human Motion Reconstruction under setting S3.

| Method | MPJRE ↓ | MPJPE ↓ | MPJVE ↓ | Hand PE ↓ | Upper PE ↓ | Lower PE ↓ | Root PE ↓ | Jitter ↓ |
|---|---|---|---|---|---|---|---|---|
| AGRoL | 2.83 | 3.80 | 17.76 | 1.62 | 1.66 | 6.90 | 3.53 | 10.08 |
| AvatarJLM | 3.14 | 3.39 | 15.75 | **0.69** | 1.48 | 6.13 | 3.04 | 5.33 |
| AvatarPoser | 2.72 | 3.37 | 21.00 | 2.12 | 1.63 | 5.87 | 2.90 | 10.24 |
| SAGE | 2.41 | 2.95 | 16.94 | 1.15 | 1.28 | 5.37 | 2.74 | 5.27 |
| RPM | 3.01 | 3.23 | 15.53 | 2.53 | 2.09 | 5.73 | 2.97 | **3.67** |
| **MotionMAR (Ours)** | **2.20** | **2.57** | **14.21** | 0.81 | **1.13** | **4.58** | **2.46** | 4.14 |

*Table 4.* Ablation results for Human Motion Reconstruction under setting S1.

| Method | MPJRE ↓ | MPJPE ↓ | MPJVE ↓ | Hand PE ↓ | Upper PE ↓ | Lower PE ↓ | Root PE ↓ | Jitter ↓ |
|---|---|---|---|---|---|---|---|---|
| MotionMAR w/o MRN | 2.47 | 3.04 | 43.9 | 0.84 | 1.32 | 6.02 | 2.70 | 42.85 |
| MotionMAR w/o SAC | 2.59 | 3.30 | 17.30 | 1.21 | 1.41 | 7.47 | 3.22 | 5.93 |
| MotionMAR w/o TMT | 2.75 | 3.41 | 19.20 | 1.09 | 1.55 | 7.93 | 3.01 | 6.27 |
| **MotionMAR (Ours)** | **2.39** | **2.82** | **16.23** | **0.83** | **1.22** | **5.13** | **2.58** | **5.17** |

this holistic design is essential for high-fidelity reconstruction.

We further implemented a SAGE variant replacing its single-scale VQ with TMT. This makes SAGE temporally hierarchy-aware. Table 6 show TMT slightly improves SAGE's reconstruction. However, it still performs weak than MotionMAR.

### 4.4. Complexity Analysis

*Table 5.* Results comparing the complexity of other human motion reconstruction methods.

| Method | Params | FLOPs | FPS |
|---|---|---|---|
| AvatarJLM | 63.81M | 0.52G | 12.91 |
| AvatarPoser | 4.12M | 0.33G | 12.40 |
| AGRoL | 7.48M | 1.00G | 54.64 |
| SAGE | 137.35M | 4.11G | 13.81 |
| RPM | 9.89M | 0.09G | 233 |
| **MotionMAR (Ours)** | 42.36M | 1.47G | 61.76 |

We further evaluate the computational efficiency of MotionMAR against state-of-the-art methods in Table 5. As observed, diffusion-based methods like SAGE incur significant computational costs (137.35M params, 4.11G FLOPs) and low inference speeds (13.81 FPS) due to their iterative denoising process. Similarly, AvatarJLM suffers from high parameter counts and limited FPS. While RPM achieves the

highest inference speed (233 FPS) with minimal parameters, its simplified architecture compromises reconstruction accuracy and structural coherence, as discussed in the quantitative results.

In contrast, MotionMAR achieves a compelling balance between model complexity and runtime performance. With 42.36M parameters and 1.47G FLOPs, our model delivers an inference speed of 61.76 FPS. This performance comfortably exceeds the standard real-time threshold (typically 30 or 60 FPS) required for VR/AR applications. These results demonstrate that MotionMAR is not only accurate but also computationally efficient enough for practical online deployment.

## 5. Conclusion

In this work, we introduce MotionMAR, a multi-scale autoregressive framework for human motion reconstruction that explicitly models the intrinsic temporal hierarchy of human kinematics. By adopting a temporal coarse-to-fine strategy, MotionMAR progressively refines motion from global low-frequency trajectories to local high-frequency dynamics, enabling accurate and coherent sequence generation. The integration of a Temporal Multi-scale VQ-VAE, a Scale-aware Control Module, and a Motion Refinement Network supports effective hierarchical representation learning and stable reconstruction under sparse observations. Extensive experiments demonstrate that MotionMAR achieves superior reconstruction accuracy, temporal consistency, and robustness, providing a principled and biologically inspired solution for motion modeling in VR/AR applications.

## Acknowledgments

This work was partially supported by the Fundamental Research Funds for the Central Universities(No. 20720230033), by Xiaomi Young Talents Program. We are grateful to the anonymous reviewer for their valuable suggestions.

## Impact Statement

This paper introduces visual autoregressive modeling into the field of human motion reconstruction from sparse observations. Therefore, any potential societal consequences or impacts related to human motion reconstruction tasks apply here, as our work introduces new ideas that enhance full-body motion reconstruction tasks with high efficiency and practicality.

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

# A. Additional Experimental Results

*Table 6.* Ablation study on incorporating Temporal Multi-scale Tokenization (TMT) into SAGE.

| Method | MPJRE ↓ | MPJPE ↓ | MPJVE ↓ | Hand PE ↓ | Upper PE ↓ | Lower PE ↓ | Root PE ↓ | Jitter ↓ |
|---|---|---|---|---|---|---|---|---|
| SAGE | 2.53 | 3.28 | 20.62 | 1.18 | 1.39 | 6.01 | 2.95 | 6.55 |
| SAGE (TMT) | 2.39 | 2.83 | 16.36 | 0.84 | 1.22 | 5.14 | 2.58 | 5.36 |
| **MotionMAR (Ours)** | **2.39** | **2.82** | **16.23** | **0.83** | **1.22** | **5.13** | **2.58** | **5.17** |

*Table 7.* Comparison results on the motion-controller subset of the GORP dataset. Lower values indicate better performance.

| Method | MPJPE ↓ | MPJVE ↓ |
|---|---|---|
| AvatarPoser (Jiang et al., 2022) | 6.49 | 14.72 |
| AGRoL (Du et al., 2023) | 6.14 | 39.14 |
| EgoPoser (Jiang et al., 2024) | 7.21 | 15.00 |
| SAGE (Feng et al., 2024) | 6.49 | 17.84 |
| AvatarJLM (Zheng et al., 2023) | 6.07 | 10.79 |
| HMD-Poser (Dai et al., 2024a) | 6.84 | 15.13 |
| RPM (Barquero et al., 2025) | 6.83 | 10.93 |
| **MotionMAR (Ours)** | **5.98** | **10.54** |

## A.1. Limitation

As illustrated in Figure 6, both existing state-of-the-art methods and our MotionMAR encounter significant challenges in extreme scenarios: (1) unconventional movements, such as a backflip executed with the head oriented downward and legs upward, and (2) cases with minimal hand motion, which make it difficult to accurately resolve arm-crossing configurations.

In addition, although our Motion Refinement Network (MRN) is designed to correct physically implausible poses, reduce jitter, and pull joints back toward anatomically valid configurations through residual refinement, it should not be regarded as an explicit physics model. In our framework, physical plausibility is encouraged by the coarse-to-fine generation pipeline together with geometric and kinematic objectives, which improve correctness, smoothness, and temporal consistency. However, these constraints remain data-driven regularizers rather than principled physical simulation. As a result, our method can still fail in rare motions or highly ambiguous cases where stronger physics-based priors may be required.

## A.2. Evaluation on GORP dataset

AMASS remains the most comprehensive benchmark for sparse human motion reconstruction and has been widely adopted by prior work such (Feng et al., 2024; Du et al., 2023; Barquero et al., 2025). It contains over 40 hours of motion data spanning 344 subjects and 11,265 motion sequences, which makes it a strong testbed for evaluating reconstruction quality under sparse observations.

To broaden our evaluation beyond AMASS, we additionally conduct experiments on the motion-controller subset of the GORP dataset introduced by RPM (Barquero et al., 2025). This subset contains more than 14 hours of real VR gameplay data collected from 28 subjects, providing a complementary benchmark with realistic tracking noise and interaction patterns. The results of Table 7 further demonstrate that MotionMAR remains competitive on this more practical dataset setting.

At the same time, we acknowledge that other datasets, such as EgoBody (Zhang et al., 2022) and GIMO (Zheng et al., 2022), are also valuable benchmarks for studying egocentric human motion and embodied behavior. We leave a more comprehensive investigation on these datasets to future work, partly due to current dataset access and evaluation protocol limitations.

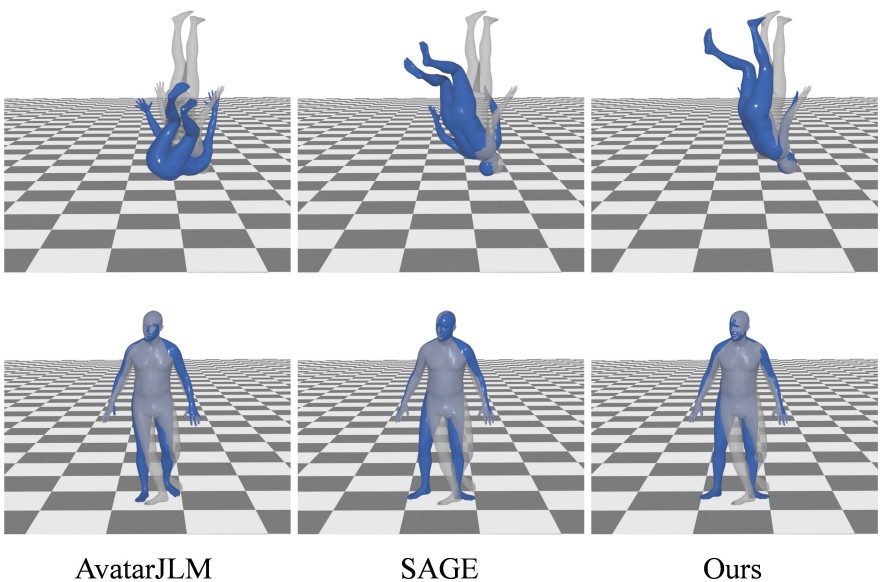

|  | AvatarJLM | SAGE | Ours |

*Figure 6.* Visualization of representative failure cases under setting S1.

*Table 8.* Ablation results with different temporal scale settings.

| Method | MPJRE ↓ | MPJPE ↓ | MPJVE ↓ | Hand PE ↓ | Upper PE ↓ | Lower PE ↓ | Root PE ↓ | Jitter ↓ | FPS ↑ |
|---|---|---|---|---|---|---|---|---|---|
| MotionMAR (Two Scales) | 2.47 | 2.93 | 17.07 | 0.87 | 1.26 | 5.33 | 2.70 | 5.51 | **91.27** |
| MotionMAR (Ours) | 2.39 | 2.82 | 16.23 | 0.83 | 1.22 | 5.13 | 2.58 | 5.17 | 61.76 |
| MotionMAR (Four Scales) | **2.36** | **2.79** | **16.01** | **0.81** | **1.19** | **5.08** | **2.53** | **5.09** | 50.88 |

*Table 9.* Analysis of spatial multi-scale modeling. We compare HiPART and a spatial variant of MotionMAR that replaces temporal coarse-to-fine generation with a spatial hierarchy.

| Method | MPJRE ↓ | MPJPE ↓ | MPJVE ↓ | Hand PE ↓ | Upper PE ↓ | Lower PE ↓ | Root PE ↓ | Jitter ↓ |
|---|---|---|---|---|---|---|---|---|
| HiPART | 2.75 | 3.71 | 98.19 | 1.74 | 1.93 | 8.54 | 3.16 | 107.7 |
| MotionMAR (Spatial) | 2.39 | 3.02 | 19.56 | 1.01 | 1.58 | 6.91 | 2.83 | 23.98 |
| **MotionMAR (Ours)** | **2.39** | **2.82** | **16.23** | **0.83** | **1.22** | **5.13** | **2.58** | **5.17** |

### A.3. Ablation results on temporal scales

To further examine the effect of temporal hierarchy design, we conduct an additional ablation study by training MotionMAR with 2, 3, and 4 temporal scales. Increasing the number of scales can potentially improve motion fidelity, but it also introduces higher computational cost and model complexity.

The results are consistent with our design intuition: increasing the hierarchy from two scales to three scales leads to clear gains in reconstructing high-frequency motion details, whereas introducing a fourth scale yields only marginal improvements. Therefore, using three temporal scales (Coarse, Mid, Fine) offers the best trade-off between reconstruction quality and computational efficiency.

### A.4. Analysis of Spatial Multi-Scale Modeling

While our proposed MotionMAR framework primarily leverages a temporal multi-scale strategy, we also investigate the efficacy of spatial multi-scale modeling for sparse motion reconstruction. To comprehensively evaluate the impact of spatial hierarchies, we introduce a new variant, denoted as MotionMAR(Spatial).

Center to Periphery

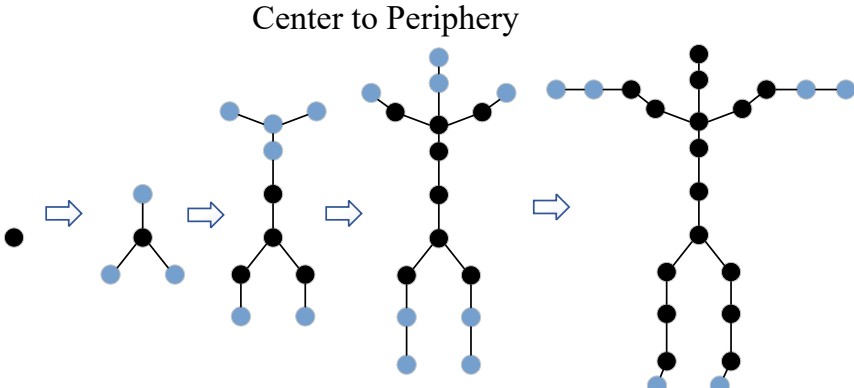

*Figure 7.* We developed a new MotionMAR variant, MotionMAR(Spatial). It replaces the temporal strategy with a spatial hierarchy that generates the skeleton from core to extremities (root, torso, limbs, then hands).

Instead of employing the temporal coarse-to-fine strategy, MotionMAR(Spatial) utilizes a spatial hierarchical structure that generates the human skeleton progressively from the core to the extremities (i.e., root → torso → limbs → hands). The detailed architecture of this spatial variant is illustrated in Figure 7.

We evaluate MotionMAR(Spatial) alongside other spatial multi-scale methods, such as HiPART (previously evaluated in Table 1). The quantitative results are detailed in Table 9. The comparison reveals that baselines relying on spatial hierarchies—including both HiPART and our MotionMAR(Spatial) variant—yield suboptimal performance compared to our temporal multi-scale approach. Specifically, the spatial hierarchical generation exhibits noticeable limitations in mitigating joint jitter and preserving global motion coherence. These findings further validate our design choice of prioritizing temporal scales to ensure stable and coherent motion reconstruction.

Considering both multi-scale spatial and temporal relations could lead to a high computational burden, hindering real-time deployment. Our temporal-only design encodes temporal topology holistically, bounding total self-attention cost across three scales $(T/4, T/2, T)$ to $\frac{21}{16}T^2d$. Conversely, decoupling both temporally and spatially with $J = 22$ joints at the finest scale skyrockets complexity to $O(T^2J^2d)$. This $\sim 369\times$ increase ($\frac{16J^2}{21}$) in core FLOPs precludes real-time VR/AR inference. Thus, we maintain a multi-scale topology, relying on the MRN for local refinement and leaving efficient joint spatio-temporal modeling as an open challenge.

### A.5. Effectiveness of Discrete Motion Tokens

Recent work in human motion generation has increasingly adopted VQ-VAE-based discrete representations due to their strong empirical performance in motion modeling (Feng et al., 2024; Lucas et al., 2022; Jiang et al., 2023; Zhang et al., 2023). This trend suggests that discretizing motion into compact tokens can provide a more structured representation space for generative modeling.

This observation is also consistent with the quantitative results in Table 1. AvatarPoser and AvatarJLM are both based on continuous regression paradigms, whereas MotionMAR and SAGE rely on discrete representations. As shown by the comparison, the methods built on discrete motion tokens achieve clearly better reconstruction quality and are more effective at preserving realistic full-body dynamics than continuous regression baselines.

To further verify the importance of discrete representations, we design an additional variant, MotionMAR (Continuous), by replacing the VQ-VAE in MotionMAR with a continuous autoencoder. Under this setting, the autoregressive network no longer predicts discrete token indices with a cross-entropy objective; instead, it directly regresses continuous latent vectors using an MSE loss. The results in Table 10 show that this continuous variant performs substantially worse than the original MotionMAR. These results confirm that modeling human motion with discrete tokens learned by VQ-VAE leads to higher motion fidelity and lower reconstruction error than using continuous latent regression.

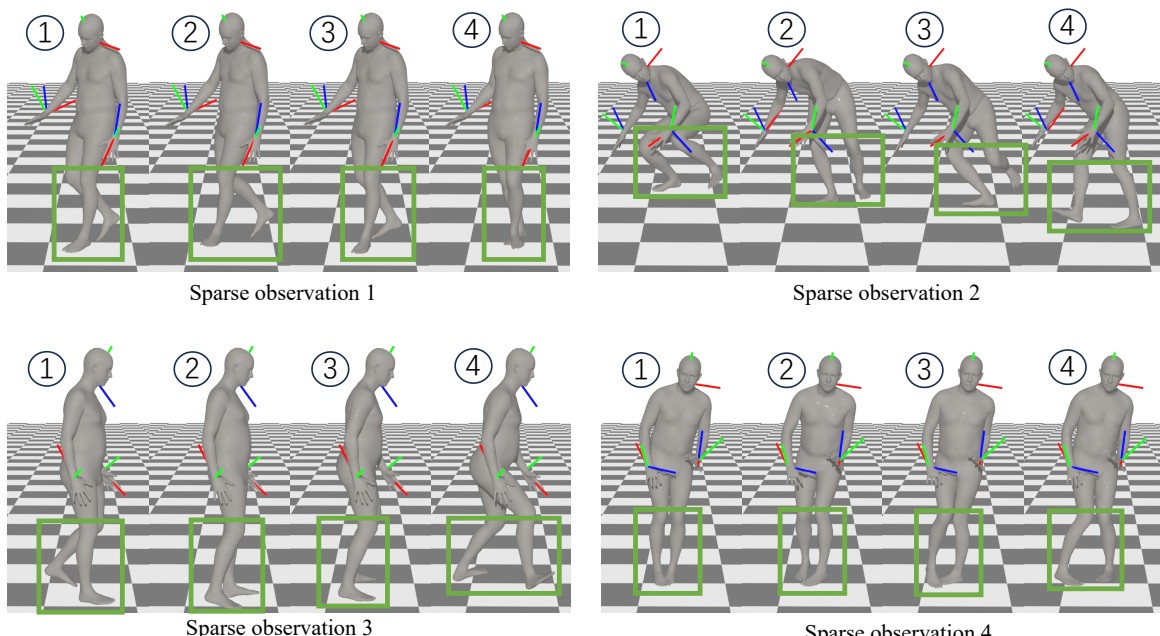

*Figure 8.* Multimodal pose generation. Our model generates four diverse, kinematically valid full-body configurations from the exact same sparse observation input, effectively avoiding deterministic mode collapse.

*Table 10.* Ablation results with continuous motion representation.

| Method | MPJRE ↓ | MPJPE ↓ | MPJVE ↓ | Hand PE ↓ | Upper PE ↓ | Lower PE ↓ | Root PE ↓ | Jitter ↓ |
|---|---|---|---|---|---|---|---|---|
| SAGE | 2.53 | 3.28 | 20.62 | 1.18 | 1.39 | 6.01 | 2.95 | 6.55 |
| MotionMAR (Continuous) | 2.53 | 3.37 | 26.01 | 1.62 | 1.49 | 6.10 | 2.97 | 11.67 |
| **MotionMAR** | **2.39** | **2.82** | **16.23** | **0.83** | **1.22** | **5.13** | **2.58** | **5.17** |

## A.6. Multimodal Full-Body Reconstruction from Sparse Observations

Reconstructing full-body motion from sparse head and hand tracking is a well-established task in the literature on sparse-observation human motion reconstruction (Feng et al., 2024; Du et al., 2023; Barquero et al., 2025). Although the inputs are sparse, the continuous trajectories of the head and hands still provide rich kinematic cues about global motion tendency, body balance, and temporal coordination. Building on these observations, our model leverages data-driven priors learned from large-scale motion datasets to infer anatomically plausible full-body poses that are consistent with the sparse controls.

We also respectfully clarify that our model does not collapse to a single "same predicted shape" for ambiguous lower-body configurations. Unlike deterministic continuous regression models, MotionMAR adopts a discrete autoregressive formulation that predicts a probability distribution over motion tokens at each generation stage. This probabilistic token modeling provides a natural mechanism for representing ambiguity in plausible lower-body states under the same sparse upper-body observations.

By sampling from the predicted token distribution, the model can generate multiple valid full-body reconstructions, including different yet plausible leg configurations that remain consistent with the observed head and hand motion. In this sense, the proposed discrete autoregressive framework explicitly mitigates deterministic mode collapse and is better suited to modeling

*Table 11.* Ablation study on replacing the bidirectional GRU in the Motion Refinement Network with a causal (unidirectional) GRU.

| Method | MPJRE ↓ | MPJPE ↓ | MPJVE ↓ | Hand PE ↓ | Upper PE ↓ | Lower PE ↓ | Root PE ↓ | Jitter ↓ | FPS ↑ |
|---|---|---|---|---|---|---|---|---|---|
| SAGE | 2.53 | 3.28 | 20.62 | 1.18 | 1.39 | 6.01 | 2.95 | 6.55 | 13.81 |
| MotionMAR (Causal GRU) | 2.45 | 2.97 | 20.42 | 0.95 | 1.31 | 5.39 | 2.70 | 6.25 | **64.25** |
| **MotionMAR (Ours)** | **2.39** | **2.82** | **16.23** | **0.83** | **1.22** | **5.13** | **2.58** | **5.17** | 61.76 |

multimodal pose uncertainty. Representative qualitative results are shown in Figure 8, where MotionMAR consistently produces anatomically plausible lower-body reconstructions under sparse observations.

### A.7. Ablation study in the Motion Refinement Network

To further evaluate the design of the Motion Refinement Network, we replace the bidirectional GRU with a causal (unidirectional) GRU and compare the resulting variant with both SAGE and the original MotionMAR. As shown in Table 11, using a causal GRU leads to a slight degradation in motion smoothness, which is expected because the model no longer has access to future-frame context during refinement. Nevertheless, this variant still performs better than SAGE on most reconstruction metrics. It also runs slightly faster than the original MotionMAR, although this speed gain comes with reduced generation quality.

*Table 12.* Design choice for temporal resampling. We compare the proposed latent-space interpolation strategy with a variant that directly samples the raw sparse tracking signals at multiple temporal scales.

| Method | MPJRE ↓ | MPJPE ↓ | MPJVE ↓ | Hand PE ↓ | Upper PE ↓ | Lower PE ↓ | Root PE ↓ | Jitter ↓ |
|---|---|---|---|---|---|---|---|---|
| SAGE | 2.53 | 3.28 | 20.62 | 1.18 | 1.39 | 6.01 | 2.95 | 6.55 |
| MotionMAR (Directly Sample) | 2.42 | 2.88 | 16.73 | 0.84 | 1.25 | 5.22 | 2.65 | 5.41 |
| **MotionMAR (Ours)** | **2.39** | **2.82** | **16.23** | **0.83** | **1.22** | **5.13** | **2.58** | **5.17** |

*Table 13.* Ablation results with the geodesic distance on the SO(3).

| Method | MPJRE ↓ | MPJPE ↓ | MPJVE ↓ | Hand PE ↓ | Upper PE ↓ | Lower PE ↓ | Root PE ↓ | Jitter ↓ |
|---|---|---|---|---|---|---|---|---|
| SAGE | 2.53 | 3.28 | 20.62 | 1.18 | 1.39 | 6.01 | 2.95 | 6.55 |
| MotionMAR (SO3) | 2.39 | 2.83 | 16.36 | 0.84 | 1.22 | 5.14 | 2.58 | 5.36 |
| **MotionMAR** | **2.39** | **2.82** | **16.23** | **0.83** | **1.22** | **5.13** | **2.58** | **5.17** |

### A.8. Design Choice for Temporal Resampling

We evaluate our design choice of interpolating extracted features against a baseline that directly samples the raw input in Scale-aware Control Module. Temporally downsampling the sparse raw tracking signals prior to feature extraction runs the risk of discarding critical high-frequency motions. By extracting features from the full-resolution input, our encoder captures the complete kinematic context, and resampling within the dense latent space preserves these priors more effectively. We compare our approach with MotionMAR (Directly sample), a variant that directly samples the original tracking signals at different temporal scales. The quantitative results in Table 12 demonstrate that our latent interpolation method yields superior performance, confirming its effectiveness in preserving motion details.

## B. Network Architecture Details

### B.1. Temporal Multi-scale VQ-VAE

Algorithm 1 first encodes the input motion into $H$, and then performs multi-scale quantization to obtain tokens at different temporal resolutions $R(t_k)$. Algorithm 2 retrieves tokens $R$ from the codebook. To align the discrete latent $Z$ with the sparse observations, we first embed the sparse observations $X$, concatenate them with $Z$, and feed the result into the decoder to produce the reconstructed motion $\hat{\theta}$. Here, $\phi_k$ denotes the residual module used in the temporal multi-scale tokenization process.

### B.2. Motion Refinement Network

Next, we will introduce more details about the Motion Refinement Network. To ensure training stability and preserve the initial structural integrity, we initialize the weights of the final linear projection layer with a near-zero gain (set to 0.01). This initialization strategy ensures that the network behaves as an identity mapping at the start of training, allowing it to progressively learn precise refinements without disrupting the global trajectory established by the autoregressive stage. To

---

**Algorithm 1** Temporal Multi-scale VQ-VAE Encoding

---

**Require:** Human motion pose $\theta$;
1: **Hyperparameters:** steps $K$, resolutions $(t_k)_{k=1}^{K}$;
2: $H = E(\theta), \ R = [\ ]$;
3: **for** $k = 1, \ldots, K$ **do**
4:    $q_k = Q_k(\text{interpolate}(H, t_k))$;
5:    $R = \text{queue\_push}(R, q_k)$;
6:    $z_k = \text{lookup}(\mathcal{Z}, q_k)$;
7:    $z_k = \text{interpolate}(z_k, t_k)$;
8:    $H = H - \phi_k(z_k)$;
9: **end for**
10: **Return** multi-scale tokens $R$;

---

**Algorithm 2** Temporal Multi-scale VQ-VAE Decoding

---

**Require:** multi-scale tokens $R$, Sparse observations $X_t$;
1: **Hyperparameters:** steps $K$, resolutions $(t_k)_{k=1}^{K}$;
2: $Z = 0$;
3: **for** $k = 1, \ldots, K$ **do**
4:    $q_k = \text{queue\_pop}(R)$;
5:    $z_k = \text{lookup}(\mathcal{Z}, q_k)$;
6:    $z_k = \text{interpolate}(z_k, t_k)$;
7:    $Z = Z + \phi_k(z_k)$;
8: **end for**
9: $X^{'} = \text{Embedding}(X_t)$;
10: $\hat{\theta} = D(\text{Concat}(Z, X^{'}))$;
11: **Return** reconstructed human motion $\hat{\theta}$;

---

guide the Motion Refinement Network in generating physically plausible and geometrically accurate motions, we impose two specific constraints during training:

**Rotation Loss** ($\mathcal{L}_{rot}$): To ensure high-fidelity pose reconstruction, we compute the loss in the interpretable axis-angle space rather than the latent 6D space. Let $\mathbf{R}_{pred}$ and $\mathbf{R}_{gt}$ denote the predicted and ground-truth rotations converted to axis-angle representation. The loss is defined as the mean absolute angular error:

$$\mathcal{L}_{rot} = \frac{1}{T \cdot J} \sum_{t=1}^{T} \sum_{j=1}^{J} \|\text{wrap}(\mathbf{R}_{pred}^{t,j} - \mathbf{R}_{gt}^{t,j})\|_1, \tag{9}$$

where $J$ is the number of joints, and $\text{wrap}(.)$ denotes the operation that maps the angular difference into the range $[-\pi, \pi]$ (implemented via matrix-to-axis-angle conversion). Using the $L_1$ metric in axis-angle space is a common practice in human motion reconstruction from sparse observations, and related methods such as SAGE, AGRoL, and RPM also optimize motion quality in comparable rotation spaces (Feng et al., 2024; Du et al., 2023; Barquero et al., 2025). At the same time, we agree that the geodesic distance on $SO(3)$ is a mathematically principled alternative for measuring rotational discrepancy. To examine this design choice, we conduct an additional ablation by replacing the loss in Equation (9) with an $SO(3)$ geodesic-distance loss. As shown in Table 13, the current $L_1$ formulation yields slightly better overall performance than the $SO(3)$ variant, and we therefore retain it in the final model.

**Multi-scale Velocity Loss** ($\mathcal{L}_{vel}$): To guarantee temporal smoothness and suppress jitter, we apply constraints on the motion trajectories at multiple temporal strides. The velocity loss is formulated as:

$$\begin{aligned} \mathcal{L}_{\text{vel}} = &\|(\theta_{gt,t} - \theta_{gt,t-1}) - (\hat{\theta}_{\text{final},t} - \hat{\theta}_{\text{final},t-1})\|_1 \\ &+ \|(\theta_{gt,t} - \theta_{gt,t-3}) - (\hat{\theta}_{\text{final},t} - \hat{\theta}_{\text{final},t-3})\|_1, \end{aligned} \tag{10}$$

---

**Algorithm 3** Training Motion Autoregressive Network and Motion Refinement Network

---

**Require:** Ground-truth motion $\theta_{gt}$ and sparse observations $X$;
**Require:** Temporal Multi-scale VQ-VAE encoder $E$ and quantizers $\{Q_k\}_{k=1}^K$ with codebook $\mathcal{Z}$;
**Require:** Motion Autoregressive Network (MAN) with parameters $\psi$; Motion Refinement Network (MRN) with parameters $\omega$.

1: Initialize MAN parameters $\psi$ and MRN parameters $\omega$.
2: **for** each mini-batch $(\theta_{gt}, X)$ **do**
3:   Obtain ground-truth multi-scale tokens $R^{gt} = \left(q_1^{gt}, \ldots, q_K^{gt}\right)$ by encoding $\theta_{gt}$ with the Temporal Multi-scale VQ-VAE (Algorithm 1).
4:   Construct scale-aligned control features $\{C_k\}_{k=1}^K$ from $X$ (Scale-aware Control Module).
5:   $\mathcal{L}_{MAN} \leftarrow 0$
6:   **for** $k = 1, \ldots, K$ **do**
7:     **Teacher forcing:** use ground-truth tokens from previous scales $q_{<k}^{gt}$ as input context.
8:     Predict token distribution $p_\psi(q_k \mid q_{<k}^{gt}, C_k)$ with MAN.
9:     $\mathcal{L}_{MAN} \mathrel{+}= \text{CE}\!\left(p_\psi(q_k \mid q_{<k}^{gt}, C_k),\ q_k^{gt}\right)$.
10:   **end for**
11:   Update $\psi$ by descending gradient of $\mathcal{L}_{MAN}$.
12:   Decode motion from ground-truth tokens $R^{gt}$ with the VQ-VAE decoder to obtain an initial reconstruction $\hat{\theta}$ (Algorithm 2).
13:   Refine $\hat{\theta}$ with MRN to obtain $\hat{\theta}_{final} = \hat{\theta} + \text{MRN}_\omega(\hat{\theta})$.
14:   Compute refinement loss $\mathcal{L}_{ref} = \mathcal{L}_{rot} + \mathcal{L}_{vel}$ (as defined in Section 3.2.3).
15:   Update $\omega$ by descending gradient of $\mathcal{L}_{ref}$.
16: **end for**
**Ensure:** Trained MAN parameters $\psi$ and MRN parameters $\omega$.

---

The first term imposes constraints on immediate frame-to-frame transitions to eliminate high-frequency flickering, while the second term (t-3) regularizes short-term dynamics to prevent temporal drift.

## C. Implementation and Training Details

### C.1. Hyperparameter Settings

For the human motion reconstruction task, we define the number of hierarchical scales as $K = 3$, with corresponding temporal resolutions set to $t_1 = 5$, $t_2 = 10$, and $t_3 = 20$ frames, respectively. The Temporal Multi-scale VQ-VAE takes continuous human motion sequences with a fixed length of $T = 20$ as input, and its encoder-decoder architecture is designed to preserve the temporal dimension, ensuring the output resolution matches the input. Similarly, the sparse observation signals fed into the Motion Autoregressive Network (MAN) are segmented into continuous sequences of the same length ($T = 20$). Regarding the network architecture, the MAN employs a depth of 8 layers (comprising Transformer blocks with Adaptive Layer Normalization) for each autoregressive scale. All training and evaluation experiments were conducted on a single NVIDIA RTX 4090 GPU.

To ensure complete reproducibility, the final experimental configurations, along with comprehensive details regarding the architectural setups and mathematical workflows, are provided in Table 14.

### C.2. Training Strategy

Algorithm 3 outlines the training procedure for the generative and refinement components. We assume the Temporal Multi-scale VQ-VAE is pre-trained and frozen. In each training iteration, we first map the ground-truth motion $\theta_{gt}$ into ground-truth multi-scale tokens $R^{gt}$ using the VQ-VAE encoder. Simultaneously, the sparse observations $X$ are processed into scale-aligned control features $\{C_k\}_{k=1}^K$. The training is divided into two phases:

1. **MAN Optimization:** We employ a **teacher forcing** strategy. For each scale $k$, the MAN predicts the current token distribution conditioned on the ground-truth tokens from previous scales ($q_{<k}^{gt}$) and the corresponding control feature $C_k$. The network parameters $\psi$ are updated by minimizing the Cross-Entropy loss ($\mathcal{L}_{MAN}$).

*Table 14.* Architectural setups, optimization details, and loss configurations for the main components of MotionMAR.

| Module | Architectural Design | Optimization & Hyperparameters | Loss Functions & Weights |
|---|---|---|---|
| Temporal Multi-Scale (TMT) VQ-VAE | 4-layer Transformer encoder/decoder, hidden dimension 512, 4 heads, dropout 0.1; 2-layer MLP projection; shared codebook with size 1024 and dimension 256; temporal scales 5, 10, 20 | AdamW; learning rate $1e-4$; weight decay $1e-4$; batch size 512; 60 epochs | VQ loss: 0.25; reconstruction loss: 1.0; local joint loss: 5.0; hand pose loss: 5.0 |
| Scale-aware Control (SAC) Module | Linear layer with hidden dimension 1024; Conv1d-ReLU-Conv1d with kernel size 3, stride 1, padding 1; 1D interpolation to match VQ scales | Optimized end-to-end with MAN | Supervised by MAN cross-entropy loss |
| Motion Autoregressive Network (MAN) | 8 and 16-layer causal Transformer, hidden dimension 1024, 16 heads; AdaLNSelfAttn and cross-scale causal mask; global AdaLN and AdaLN-Linear head | AdamW; learning rate $1e-4$; weight decay $1e-4$; batch size 512; 500 epochs; AdaLN-$\gamma$: $1e-3$; condition dropout 0.1 | Cross-Entropy loss |
| Motion Refinement Network (MRN) | 2-layer BiGRU, hidden dimension 512, dropout 0.1; linear layer for spatial residual deltas; smooths 132-dimensional continuous pose | AdamW; learning rate $1e-4$; weight decay $1e-4$; batch size 1; 200 epochs | Velocity loss $v_1$: 60.0; velocity loss $v_2$: 20.0; rotation loss: 0.1 |

2. **MRN Optimization:** We decode the ground-truth tokens back into a continuous motion $\hat{\theta}$ via the VQ-VAE decoder. The MRN then predicts a residual to refine $\hat{\theta}$. The parameters $\omega$ are updated by minimizing the geometric and kinematic losses ($\mathcal{L}_{ref}$), ensuring the final output is smooth and physically plausible.

