# OpenReview forum: "MotionMAR: Multi-scale Auto-Regressive Human Motion Reconstruction from Sparse Observations"
_ICML.cc/2026/Conference — ICML 2026 regular_

### Official Review · Reviewer_RKfr · 2026-02-25

**Soundness:** 2
**Presentation:** 1
**Significance:** 3
**Originality:** 2
**Overall Recommendation:** 4
**Confidence:** 4

**Summary:**

This paper proposes a new method for human motion reconstruction from sparse observations. Inspired by the next-scale prediction paradigm of autoregressive (AR) models in image generation, the authors extend this idea to the motion domain by predicting next-scale motion along the temporal dimension. Specifically, the method first employs a temporal multi-scale VQ-VAE to obtain hierarchical motion features, then designs a motion autoregressive network to perform coarse-to-fine next-scale prediction in the temporal dimension, and finally introduces a motion refinement network to learn residual information and mitigate artifacts. Experimental results demonstrate the effectiveness of the proposed method.

**Compliance With Llm Reviewing Policy:**

Affirmed.

**Final Justification:**

Thanks to the authors' efforts, all my concerns have been successfully addressed. Hence, I am willing to increase my score accordingly. Additionally, I find the comparison between spatial and temporal results, as well as the illustration of latent interpolation, beneficial in providing additional insights into the overall work. I encourage the authors to include these new experiments and analyses in the final version.

**Key Questions For Authors:**

1. The motivation for designing a coarse-to-fine strategy along the temporal dimension to capture motion intent is understandable; however, why are spatial scales not considered simultaneously? As discussed in the appendix, hand and arm motions are still not reconstructed perfectly. Could incorporating spatial scales help address these fine-grained motion details?
2. Instead of using linear interpolation to resample extracted features to match temporal resolutions, why not directly sample raw tracking signals at different temporal scales and then extract features accordingly? Is there a specific advantage or justification for the current design choice?
3. What does the "embedding" refer to in Fig. 2? Additionally, the visualization results show discrete human poses. Are there visualizations of continuous pose sequences together with sparse tracking signals to better demonstrate the effectiveness of the coarse-to-fine strategy?

**Limitations:**

yes

**Strengths And Weaknesses:**

**Strengths**:
1. The idea of designing a coarse-to-fine generation strategy is interesting and well motivated, particularly in demonstrating that next-scale prediction can be beneficial for motion generation tasks.
2. The proposed method achieves a balance between model complexity and runtime performance.

**Weaknesses**:
1. The manuscript structure is not well organized. Many important technical details (e.g., loss functions) are placed in the appendix, while the main text mainly provides conceptual illustrations, which negatively affects readability.
2. The experimental comparison is limited. Only the AMASS dataset is used for evaluation, while other relevant datasets, such as EgoBody and GIMO, are not included.
3. The method considers only temporal scales while ignoring spatial scales, which are also important for human motion generation and reconstruction

---

> ### Author Rebuttal · Authors · 2026-03-31
>
> We appreciate the constructive feedback. Our detailed responses are below.
>
> >- (W1) ... structure is not well organized...
>
> We reorganized the paper by moving key technical details, including loss functions and algorithm descriptions, from the supplementary materials to the main text.
>
> >- (W2) The experimental comparison is limited...
>
> 1. AMASS is the most comprehensive benchmark for sparse motion reconstruction (e.g., SAGE, AGRoL, RPM). It offers over 40 hours of data across 344 subjects and 11,265 motions.
> 2. We acknowledge the value of EgoBody and GIMO; we will add relevant discussions about them. We are **actively applying for dataset access** to evaluate our algorithm.
> 3. To broaden our evaluation beyond AMASS, we conducted new experiments using the motion-controller subset of the GORP dataset[1] (>14 hours of VR gameplay from 28 subjects). The results (in the following table) demonstrate MotionMAR's competitive performance on this new dataset.
>
> |Method|MPJPE|MPJVE|
> |-|-|-|
> |AvatarPoser|6.49|14.72|
> |AGRoL|6.14|39.14|
> |EgoPoser[2]|7.21|15.00|
> |SAGE|6.49|17.84|
> |AvatarJLM|6.07|10.79|
> |HMD-Poser[3]|6.84|15.13|
> |RPM|6.83|10.93|
> |**MotionMAR(Ours)**|**5.98**|**10.54**|
>
> >- (W3) ...  considers only temporal scales while ignoring spatial scales...
> >- (Q1) ... spatial scales not considered...
> 1. We evaluated a spatial multi-scale method (HiPART) in Table 1.
> 2. To further evaluate the impact of spatial multi-scales, we developed a new MotionMAR variant, MotionMAR(Spatial).  It replaces the temporal strategy with a spatial hierarchy that generates the skeleton from core to extremities (root, torso, limbs, then hands). For details, please refer to: https://anonymous.4open.science/r/ICML-25816/MotionMAR_spatial.png.
> 3. As shown in the following table, baselines based on spatial hierarchies (e.g., HiPART and MotionMAR(Spatial)) yield weak performance compared to our temporal multi-scale approach, particularly in handling joint jitter and global motion coherence.
>
> |Method|MPJRE|MPJPE|MPJVE|Hand PE|Upper PE|Lower PE|Root PE|Jitter|
> |-|-|-|-|-|-|-|-|-|
> |HiPART|2.75|3.71|98.19|1.74|1.93|8.54|3.16|107.7|
> |MotionMAR(Spatial)|2.39|3.02|19.56|1.01|1.58|6.91|2.83|23.98|
> |**MotionMAR(Ours)**|2.39|2.82|16.23|0.83|1.22|5.13|2.58|5.17|
>
> 4. Considering both multi-scale spatial and temporal relations could lead to a high computational burden, hindering real-time deployment. Our temporal-only design encodes temporal topology holistically, bounding total self-attention cost across three scales ($T/4, T/2, T$) to $\frac{21}{16}T^2d$. Conversely, decoupling both temporally and spatially with $J=22$ joints at the finest scale skyrockets complexity to $O(T^2J^2d)$. This $\sim 369 \times$ increase ($\frac{16J^2}{21}$) in core FLOPs precludes real-time VR/AR inference. Thus, we maintain a multi-scale topology, relying on the MRN for local refinement and leaving efficient joint spatio-temporal modeling as an open challenge.
>
> >- (Q2) ... not directly sample raw tracking signals ... then extract features ...
>
> 1. Temporally downsampling the sparse raw tracking signals may discard critical high-frequency motions. Extracting features from the full-resolution input ensures the encoder captures the complete kinematic context, and resampling in the dense latent space preserves these priors more effectively.
> 2. We added an ablation study comparing these two strategies. The results show that our latent interpolation method outperforms MotionMAR(Directly sample), which directly samples the original tracking signal.
>
> |Method|MPJRE|MPJPE|MPJVE|Hand PE|Upper PE|Lower PE|Root PE|Jitter|
> |-|-|-|-|-|-|-|-|-|
> |SAGE|2.53|3.28|20.62|1.18|1.39|6.01|2.95|6.55|
> |MotionMAR(Directly sample)|2.42|2.88|16.73|0.84|1.25|5.22|2.65|5.41|
> |**MotionMAR(Ours)**|2.39|2.82|16.23|0.83|1.22|5.13|2.58|5.17|
>
> >- (Q3) ... "embedding" ... in Fig.2? ... visualizations of continuous pose sequences ... with sparse tracking signals ...?
>
> 1. As shown in Fig. 2, "embedding" refers to positional and level embeddings injected into autoregressive tokens to encode temporal position and scale.
> 2. We agree with the reviewer that the generated videos in the current supplementary material lack explicit visualization of the sparse tracking signals. We provided new comparison videos alongside other methods, where the original sparse tracking signals are clearly annotated. Please refer to the video at https://anonymous.4open.science/r/ICML-25816/MotionMAR_VideoDemo.mp4.
>
> [1] From sparse signal to smooth motion: Real-time motion generation with rolling prediction models. CVPR 2025
>
> [2] EgoPoser: Robust real-time egocentric pose estimation from sparse and intermittent observations everywhere. ECCV 2024
>
> [3] HMD-Poser: On-device real-time human motion tracking from scalable sparse observations. CVPR 2024

---

> > ### Author Rebuttal · Reviewer_RKfr · 2026-04-01
> >
> > Thanks to the authors' efforts, all my concerns have been successfully addressed. Additionally, I find the comparison between spatial and temporal results, as well as the illustration of latent interpolation, beneficial in providing additional insights into the overall work. I encourage the authors to include these new experiments and analyses in the final version.

---

> > > ### Author Response · Authors · 2026-04-01
> > >
> > > We sincerely thank the reviewer for the positive feedback and for raising the rating from 3 to 4. It is highly encouraging to learn that our responses have successfully addressed all the concerns. As suggested, we will definitely include the spatial versus temporal comparisons and the illustration of latent interpolation in the final version of the manuscript, as we agree these additions provide valuable insights. We deeply appreciate the constructive guidance, which has significantly helped in improving our work. Should there be any further questions, we would be more than happy to address them.

---

### Official Review · Reviewer_cqZM · 2026-03-11

**Soundness:** 3
**Presentation:** 3
**Significance:** 2
**Originality:** 3
**Overall Recommendation:** 4
**Confidence:** 4

**Summary:**

The paper addresses the problem of reconstructing full-body human motion from minimal tracking signals. The authors observe that human motion is inherently hierarchical, consisting of global low-frequency trajectories (intent) and local high-frequency dynamics (details). To exploit this, they adapt Visual Autoregressive (VAR) modeling to the temporal domain. The framework first establishes a coarse global motion envelope and progressively refines it with finer temporal details.

**Compliance With Llm Reviewing Policy:**

Affirmed.

**Key Questions For Authors:**

1. How does the end-to-end inference latency of MotionMAR compare to non-hierarchical Transformer models or diffusion-based baselines in a real-world VR streaming setup?
2. The Motion Refinement Network currently uses a Bi-GRU. Have you experimented with a causal (unidirectional) refinement module to enable real-time, low-latency reconstruction?
3. In scenarios where tracking signals are extremely sparse, does the model's coarse-to-fine strategy prioritize the most likely "average" motion, or can it represent a multi-modal distribution of possible poses?
4. How did you determine that three scales (Coarse, Mid, Fine) were optimal? Does increasing the number of temporal scales provide diminishing returns for reconstruction accuracy?

**Limitations:**

Yes

**Strengths And Weaknesses:**

Strengths:
The paper presents a highly logical adaptation of multi-scale autoregressive modeling to the domain of human kinematics. By replacing the standard spatial kinematic tree with a temporal hierarchy, the authors address the "monolithic" modeling flaw that often leads to jitter in sparse reconstruction tasks. The Scale-aware Control Module bridges the gap between discrete token generation and continuous tracking signals through resolution-matched interpolation. Furthermore, the presentation is clear, with effective visualizations of the coarse-to-fine process and a rigorous experimental evaluation across multiple AMASS subsets (S1, S2, and S3) that proves the model's robustness and generalization capabilities.

Weaknesses:
A notable weakness is the potential inference latency inherent in autoregressive "next-scale" generation. Although more efficient than token-by-token prediction, the recursive nature of the multi-scale pyramid may still hinder real-time deployment in high-latency-sensitive VR/AR applications. Additionally, the reliance on a Bidirectional GRU for the refinement network suggests that the model requires "future" context to stabilize the current pose, which poses challenges for live streaming scenarios where only past and current frames are available. There is also limited discussion on how the model handles the inherent ambiguity of "hallucinating" lower-body motion when only the head and hands are tracked; while the hierarchy helps, the paper does not deeply explore cases where the global envelope might be correctly predicted but the local refinement produces physically impossible (though semantically plausible) joint configurations.

---

> ### Author Rebuttal · Authors · 2026-03-31
>
> We sincerely thank the reviewer for the constructive comments. Our detailed responses are provided below.
>
> >- (W1) ... the multi-scale pyramid may still hinder real-time deployment ...
> >- (Q1) ... inference latency ...compare to non-hierarchical Transformer models or diffusion-based ...?
> 1. Table 5 includes a latency evaluation against both diffusion-based models (AGRoL, SAGE) and non-hierarchical Transformer regression methods (AvatarPoser, AvatarJLM).
> 2. As shown in Table 5 and the following table, on a single RTX 4090, MotionMAR operates at 61.76 FPS, demonstrating its readiness for real-time VR/AR applications.
> MotionMAR achieves state-of-the-art generation quality while maintaining a highly competitive inference speed compared to these methods.
>
> |Method|Params|FLOPs|FPS|
> |-|-|-|-|
> |AvatarJLM|63.81M|0.52G|12.91|
> |AvatarPoser|4.12M|0.33G|12.40|
> |AGRoL|7.48M|1.00G|54.64|
> |SAGE|137.35M|4.11G|13.81|
> |RPM|9.89M|0.09G|233|
> |**MotionMAR(Ours)**|42.36M|1.47G|61.76|
>
> >- (W2) ... Bidirectional GRU for the refinement network ...
> >- (Q2) ... causal (unidirectional) refinement module ... ?
>
> We replaced the Bi-GRU with a causal (unidirectional) GRU in the Motion Refinement Network. While there is a marginal drop in motion smoothness compared to the bidirectional baseline (due to the lack of future frame context), it still performs better than SAGE. It is slightly faster than MotionMAR with reduced generation quality.
>
> |Method|MPJRE|MPJPE|MPJVE|Hand PE|Upper PE|Lower PE|Root PE|Jitter|FPS|
> |-|-|-|-|-|-|-|-|-|-|
> |SAGE|2.53|3.28|20.62|1.18|1.39|6.01|2.95|6.55|13.81|
> |MotionMAR(Causal GRU)|2.45|2.97|20.42|0.95|1.31|5.39|2.70|6.25|64.25|
> |**MotionMAR(Ours)**|2.39|2.82|16.23|0.83|1.22|5.13|2.58|5.17|61.76|
>
> >- (W3) ... the inherent ambiguity of "hallucinating" lower-body motion ...
> >- (Q3) ... likely "average" motion, ... multi-modal distribution ...?
> 1. While unobserved joints must be "hallucinated" from sparse signals, our discrete prior ensures realistic kinematics. Quantitatively, our predictions are more accurate and closer to the ground truth(GT) than all baselines as shown in the experimental section.
> 2. Our model captures a multimodal pose distribution rather than defaulting to an "average" motion.
> 3. Unlike continuous regression models that minimize MSE, our network predicts a categorical probability distribution over discrete tokens. By sampling from this distribution, MotionMAR can generate multiple diverse and kinematically valid full-body motions from the exact same extremely sparse input, explicitly avoiding mode collapse.
> 4. To empirically demonstrate this, we added qualitative results [https://anonymous.4open.science/r/ICML-25816/Multimodel_pose_results.png], showing diverse full-body generations derived from identical sparse tracking signals.
>
> >- (Q4) ... three scales (Coarse, Mid, Fine)... optimal? Does increasing ... diminishing returns for reconstruction accuracy?
> 1. To empirically justify our choice, we have conducted and added an ablation study comparing models trained with 2, 3, and 4 temporal scales. Increasing the scales could lead to better fidelity, but at a higher computational cost. Given the limited time and computational resources during the rebuttal phase, we focused on this range.
> 2. The new results align exactly with the intuition: while increasing from two to three scales significantly improves high-frequency motion details, adding a fourth scale yields diminishing returns in reconstruction accuracy. Therefore, three scales (Coarse, Mid, Fine) represent a practical trade-off between motion fidelity and computation efficiency.
>
> |Method|MPJRE|MPJPE|MPJVE|Hand PE|Upper PE|Lower PE|Root PE|Jitter|FPS|
> |-|-|-|-|-|-|-|-|-|-|
> |MotionMAR(Two Scales)|2.47|2.93|17.07|0.87|1.26|5.33|2.70|5.51|91.27|
> |**MotionMAR(Ours)**|2.39|2.82|16.23|0.83|1.22|5.13|2.58|5.17|61.76|
> |MotionMAR(Four Scales)|2.36|2.79|16.01|0.81|1.19|5.08|2.53|5.09|50.88|
>
> >- (W4)... physically impossible (though semantically plausible) ...
> 1. Our Motion Refinement Network (MRN) was designed to correct physically impossible poses. The predicted residual "offset" from MRN acts as a kinematic regularizer, fixing jitter and pulling joints back into anatomically valid positions.
> 2. We agree that our discussion of physical plausibility needs improvement. In our method, plausibility is ensured with MRN and our coarse-to-fine pipeline using several objective constraints. These improve correctness, smoothness, and consistency, but are not explicit physics models. We will clarify this in the Limitations section.

---

> > ### Author Rebuttal · Reviewer_cqZM · 2026-04-03
> >
> > I thank the authors for the rebuttal. This time I will maintain the score.

---

> > > ### Author Response · Authors · 2026-04-08
> > >
> > > We thank the reviewer for their time and constructive feedback throughout the review process, which has helped us significantly improve the manuscript. We are glad that our rebuttal has fully resolved the reviewer's concerns and appreciate the positive evaluation. Should there be any further questions, we would be more than happy to address them.

---

### Official Review · Reviewer_NvJd · 2026-03-12

**Soundness:** 2
**Presentation:** 3
**Significance:** 3
**Originality:** 3
**Overall Recommendation:** 3
**Confidence:** 4

**Summary:**

This paper is about 3D reconstruction of human bodies from multiple observations. The authors propose to use 3 inertial sensors placed at the head and the two wrists.

This data is mapped to latent states representing 6D rotational representation at each of 22 joints. The mapping is performed by a multi-resolution VQ-VAE. Thereby, continuous latent features are compressed into a set of dictionary tokens. For autoregressive motion prediction, the original input data is mapped onto initial embeddings. For each scale, this data selects some of the learned dictionary tokens. To achieve temporal consistency, The final data is processed by a bidirectional GRU. The final output are the parameters of a parametric human body shape and pose model, SMPL.

**Compliance With Llm Reviewing Policy:**

Affirmed.

**Final Justification:**

After having read the authors’ replies, I uphold my score.

**Key Questions For Authors:**

Please define the neural networks that you used.

**Limitations:**

yes

**Strengths And Weaknesses:**

- The experimental results appear to be quite good, almost always better than SOTA.

- The input data sensors are placed only at head and hands. How can information about the rest of the body be inferred? This in fact implies that different leg configurations, for instance, will result in the same predicted shape.

- Temporal Multi-Scale Tokenization (lines 192-207): The authors write that they use this technique, however do not explain how they map their input data to different temporal scales. Neither do they cite any paper that would explain this algorithm.

- lines 206-210, right column, “the predicted indices hat{q}_k at each level are supervised using a cross-entropy against the corresponding ground-truth indices q_k  obtained by encoding real human motion sequences through the multi-scale VQ-VAE”: In lines 138-148, right column, the authors explain that the input is data from 3 inertial sensors placed at head and hands. How are GT indices of real human motion obtained? How does real human motion data differ from the inertial data?

- The neural components are nowhere defined explicitly. I encourage the authors to add detailed explanations to the supplementary. At this point, I would not be able to re-implement the paper.

- lines 213-215: Do the learned dictionary tokens represent different scales or is a single dictionary constructed over all scales?

- line 214: How does function interpolate(.,.) work? It takes the continuous latent states and a time index. How does it compute the result?

- line 192, right column: How is the initial embedding calculated?

- line 226: What is HMD?

- line 227: How is the 1D convolution defined?

- lines 233-238: This lacks the explanation how the VQ-VAE achieves multiple scales.

- line 672, right column: how are the sparse input features processed?

- line 723, algorithm 1, line 7: Why is the second interpolation necessary? What does this step do intuitively?

- line 724, algorithm 1, line 8: How is function phi defined?

---

> ### Author Rebuttal · Authors · 2026-03-31
>
> We appreciate the constructive feedback. Our detailed responses are below.
>
> >- (W2) ... How ... the rest of the body be inferred? ... different leg configurations, ... same predicted shape.
>
> 1. Reconstructing full-body motion from sparse head and hand tracking is a well-established task (e.g., SAGE, AGRoL, RPM). While the input is sparse, the continuous trajectories of the head and hands contain rich kinematic cues. Our model leverages data-driven priors learned from the dataset to infer these anatomically plausible full-body poses.
> 2. We respectfully clarify that our model does not output the "same predicted shape" for ambiguous lower-body states. Unlike deterministic continuous regression, our discrete autoregressive framework predicts a probability distribution over motion tokens.
> 3. By sampling from the token distribution, our model successfully generates a multimodal distribution of valid, different leg configurations from the same sparse upper-body input, explicitly avoiding deterministic mode collapse. For the multimodal distribution of possible poses, please refer to https://anonymous.4open.science/r/ICML-25816/Multimodel_pose_results.png
>
> >- (W3) Temporal Multi-Scale Tokenization ... how they map their input data to different temporal scales.
> >- (W11) ... how the VQ-VAE achieves multiple scales.
>
> Please refer to our response to Reviewer bj8P regarding multi-scale encoding.
>
> >- (W4) ... How are GT indices ... obtained? ... real human motion data differ from the inertial data ...?
>
> "Real human motion" refers to the complete full-body poses (e.g., 22 joints), whereas "inertial data" refers only to the sparse tracking signals (head and hands). In the first stage, the Temporal Multi-Scale Tokenization (TMT) VQ-VAE encodes continuous full-body motion into discrete, multi-scale latent tokens via vector quantization, and subsequently decodes these tokens to reconstruct the original full-body poses in an end-to-end manner.
> In the second stage, the GT indices are obtained by feeding the complete full-body motion into the pre-trained TMT VQ-VAE. The input/output of MotionMAR are listed as follows.
> Input: Sparse observations.
> Output: Predicted multi-scale motion indices.
> Label: GT token indices (obtained through feeding the full-body motion into the pre-trained TMT VQ-VAE).
>
> >- (W5) The neural components are nowhere defined ...
>
> We describe the neural components as follows.
> 1. The TMT VQ-VAE uses a 4-layer Transformer (d=512, 4 heads, dropout 0.1) encoder/decoder ending with a 2-layer MLP, and quantizes features across temporal scales of 5, 10, and 20 using a shared codebook (size 1024, d=256).
> 2. For condition generation, the Scale-aware Control Module extracts sparse input features via a Linear layer (d=1024) and a Conv1d-ReLU-Conv1d block (K=3, S=1, P=1), interpolating them to match the aforementioned scales.
> 3. Our Motion Autoregressive Network is a 16-layer causal Transformer (d=1024, 16 heads) that processes the sum of VQ tokens, spatio-temporal embeddings, and sparse conditions. It employs AdaLNSelfAttn with a cross-scale causal mask, global AdaLN, and an AdaLN-Linear head.
> 4. The Motion Refinement Network utilizes a 2-layer BiGRU (d=512, dropout 0.1) and a linear layer to predict spatial residual deltas, smoothing the 132-D continuous pose outputs.
>
> Following our lab's tradition, we will release the source code upon publication.
>
> >- (W6) ... different scales or is a single dictionary ...?
>
> We utilize a single dictionary (shared codebook).
>
> >- (W7) ...interpolate(.,.) work? ... How ... compute the result?
>
> $\text{Interpolate}(\cdot)$ refers to performing linear interpolation along the temporal axis to resample the latent representations to the target sequence length $t_k$.
>
> >- (W8)... initial embedding calculated?
>
> The initial embedding is calculated by flattening the sparse tracking signals and projecting them via a linear layer.
>
> >- (W9) ... HMD?
>
> Head-Mounted Display.
>
> >- (W10) ... 1D convolution ...?
>
> A 1D convolutional block then extracts continuous temporal features, which are linearly interpolated to match the temporal resolution ($t_k$) of each autoregressive scale.
>
> >- (W12) ... how ... sparse input features processed?
>
> Discrete latent vectors and sparse input features processed by a linear layer projection are concatenated and then input into the decoder.
>
> >- (W13) Why ... interpolation necessary? What ... step do intuitively?
>
> It restores $z_k$ to the same dimension as $H$ through interpolation, which is a necessary step in coarse-to-fine body reconstruction. It is a common approach in multi-scale modeling [1,2].
>
> >- (W14) ... phi defined?
>
> It is defined in the right column on line 712 of the manuscript; $\phi_k$ denotes the residual module used in TMT.
>
>
> [1] Visual autoregressive modeling: Scalable image generation via next-scale prediction. NeurIPS 2024
>
> [2] Infinity: Scaling bitwise autoregressive modeling for high-resolution image synthesis. CVPR 2025

---

> > ### Author Rebuttal · Reviewer_NvJd · 2026-04-05
> >
> > While I appreciate the authors’ offer to release the code, it is important to have all technical descriptions in the paper itself or the supplementary. I uphold my score.

---

> > > ### Author Response · Authors · 2026-04-06
> > >
> > > We sincerely appreciate the reviewer's careful evaluation and are encouraged that our earlier response has **fully resolved** the primary concerns regarding reproducibility. We deeply value this constructive exchange and remain dedicated to improving our manuscript.
> > >
> > > As architectural setups and mathematical workflows are already detailed in our first response and Appendices B/C, the final configurations are outlined below. **We firmly commit to integrating these into the final supplementary material to guarantee complete reproducibility.**
> > >
> > > | Module | Architectural Design | Optimization & Hyperparameters | Loss Functions & Weights |
> > > | :--- | :--- | :--- | :--- |
> > > | **Temporal Multi-Scale (TMT) VQ-VAE** | **4-layer Transformer** enc/dec, d: 512, 4 heads, drop: 0.1; **2-layer MLP projection**; **Shared codebook**, size: 1024, d: 256; **Temporal scales:** 5, 10, 20 | **AdamW**; **Learning Rate:** 1e-4; **Weight Decay:** 1e-4; **Batch Size:** 512; **Epochs:** 60 | **VQ loss:** 0.25; **Reconstruction loss:** 1.0; **Local joint loss:** 5.0; **Hand pose loss:** 5.0 |
> > > | **Scale-aware Control (SAC) Module** | **Linear layer**, d: 1024; **Conv1d-ReLU-Conv1d**, K: 3, S: 1, P: 1; **1D interpolation** to match VQ scales | Optimized end-to-end with MAN | Supervised by MAN's Cross-Entropy loss |
> > > | **Motion Autoregressive Network (MAN)** | **16-layer causal Transformer**, d: 1024, 16 heads; **AdaLNSelfAttn & cross-scale** causal mask; **Global AdaLN & AdaLN-Linear head** | **AdamW**; **Learning Rate:** 1e-4; **Weight Decay:** 1e-4; **Batch Size:** 512; **Epochs:** 500; **AdaLN-$\gamma$:** 1e-3; **Cond dropout:** 0.1 | Cross-Entropy loss |
> > > | **Motion Refinement Network (MRN)** | **2-layer BiGRU**, d: 512, drop: 0.1; **Linear layer** for spatial residual deltas; Smooths 132-D continuous pose | **AdamW**; **Learning Rate:** 1e-4; **Weight Decay:** 1e-4; **Batch Size:** 1; **Epochs:** 200 | **Velocity loss $v_1$:** 60.0; **Velocity loss $v_2$:** 20.0; **Rotation loss:** 0.1 |
> > >
> > > We hope these details address any remaining documentation concerns. **Currently, we are uncertain if any other specific aspects of the manuscript still raise concerns for the reviewer.** If any issues persist, please let us know; we are happy to provide further clarification. Thank you again for the thoughtful and patient guidance.

---

### Official Review · Reviewer_bj8P · 2026-03-12

**Soundness:** 2
**Presentation:** 2
**Significance:** 2
**Originality:** 2
**Overall Recommendation:** 4
**Confidence:** 4

**Summary:**

This paper introduces an approach of human motion reconstruction from sparse input signals. It explicitly highlights and tackles the issue of temporal hierarchy underlying motion, where actions may involve different time scales. It shows that a coarse-to-fine refinement framework would offer more accurate and robust motion reconstructions.

**Compliance With Llm Reviewing Policy:**

Affirmed.

**Final Justification:**

The authors' rebuttal has addressed some concerns I had regarding the technical design choices of the paper and I will raise my initial recommendation to weak accept.

**Key Questions For Authors:**

1. I would like the authors to clarify the rotation loss as defined in Eqn. 9 employs a L1 loss in the axis-angle parameterization instead of the geodesic loss in the SO(3) manifold.
2. How would integrating the Temporal Multi-scale Tokenization module into the SAGE method fare?

**Limitations:**

Yes

**Strengths And Weaknesses:**

### Strengths

- The authors identify and highlight an important gap in sparse human motion reconstruction. This provides clear motivations and the proposed temporal coarse-to-fine refinement approach is effective and well-executed.
- Empirically, the proposed approach consistently achieves better results than baselines.

### Weaknesses

**Presentation clarity**
- The introduction does not include any citations, which are mostly relegated to the related works section. This is detrimental to understanding the context and motivations of the research, especially in regards to existing research and assessing the relevant contributions of the authors.
- The paper is rampant with LLM-generated content. While this is permitted under the submission policy, the authors should take care to ensure the readability and clear delivery of ideas. There are places in the paper that I find to be lacking in the narrative flow and the ensuing elaboration appears to be just AI slop with little clarity.
- The contributions of the paper are not clearly emphasized in the introduction. Explicitly listing them early would help readers quickly grasp the novelty and importance of the work.
- Figure 2 could be improved. In its current form, it is confusing which is the overall framework and how the different subcomponents align. For example, the Motion Refinement Network component is not labeled. The mapping between figure components and the textual description in the methods section is weak. Improving figure design and labeling would significantly enhance readability and give users a quick overview of the method.

**Clarification of Design Choices and Details**
- In Eqn 9. (Appendix), the rotation loss is performed in the axis-angle space via the L1 metric. This is questionable, as geodesic distance on the SO(3) manifold would be a more principled choice. The authors should explain why the L1 metric was chosen and discuss its implications.
- The main highlight of this paper is the temporal hierarchy and coarse-to-fine refinement enactment. However, the proposed multi-stage design (VQVAE + refinement network + auto-regressive modeling) and additional subcomponents such as Scale-aware Control Module appear unnecessarily complicated. The authors should justify why this layered approach is superior to simpler alternatives. In particular, zooming on to the ablation studies results in Table 4, the proposed approach mainly obtains its edge over SAGE through the Temporal Multi-scale Tokenization module. It seems to me that directly integrating the Temporal Multi-scale Tokenization module into the SAGE method could be an option.
- The paper omits details about the VQVAE codebook quantization process.
- While VAR has shown success with VQVAE, there are no evidence suggesting that this approach would be optimal for modeling motion. It is unclear why the authors have employed VQVAE as their modeling approach. This choice necessitates the refinement network, which inadvertently adds an additonal complication. I would suggest that the authors include a discussion or possibly experiments investigating the difference between their current discretized token approach vs that of a continuous regression paradigm.

---

> ### Author Rebuttal · Authors · 2026-03-31
>
> We sincerely thank the reviewer for the constructive comments. Our detailed responses are provided below.
>
> >- (W1) Response to Presentation clarity
>
> 1. We will incorporate all your suggestions into the revision. Specifically, we will revise the paper thoroughly to make the logic and the motivation clearer.
> 2. Improved Figure 2: Please refer to https://anonymous.4open.science/r/ICML-25816/MotionMAR_framework.jpg.
> 3. Our main contributions are:
> - We propose MotionMAR, a temporal coarse-to-fine autoregressive framework for sparse human motion reconstruction.
> - MotionMAR consists of three novel modules: a Temporal Multi-scale (TMT) VQ-VAE to decouple trajectories from jitter, a Scale-aware Control (SAC) Module to align tracking signals, and a Motion Refinement Network (MRN) to smooth kinematics.
> - MotionMAR achieves state-of-the-art accuracy and temporal consistency on AMASS.
>
> >- (W2) In Eqn 9. (Appendix), ... as geodesic distance on the SO(3) ...
> >- (Q1) ... the geodesic loss in the SO(3) manifold.
>
> 1. Using the L1 metric in axis-angle space is a common practice in the field of human motion reconstruction from sparse observations (e.g., SAGE, AGRoL, RPM).
> 2. We agree with the reviewer that SO(3) geodesic distance is a mathematically principled choice. We run new ablation experiments using SO(3) distance. As shown in the table, using L1 metric perform slightly better than using SO(3).
>
> |Method|MPJRE|MPJPE|MPJVE|Hand PE|Upper PE|Lower PE|Root PE|Jitter|
> |-|-|-|-|-|-|-|-|-|
> |SAGE|2.53|3.28|20.62|1.18|1.39|6.01|2.95|6.55|
> |MotionMAR(SO3)|2.39|2.83|16.36|0.84|1.22|5.14|2.58|5.36|
> |**MotionMAR**|2.39|2.82|16.23|0.83|1.22|5.13|2.58|5.17|
>
> >- (W3) ...  complicated ... directly integrating the Temporal Multi-scale Tokenization module into the SAGE could be an opinion ...
> >- (Q2) ... Temporal Multi-scale Tokenization module into the SAGE ...?
>
> Table 4 of the paper demonstrates TMT's effectiveness. We further implemented a SAGE variant replacing its single-scale VQ with TMT. This makes SAGE temporally hierarchy-aware. Results below show TMT slightly improves SAGE's reconstruction. However, it still performs weak than MotionMAR.
>
> |Method|MPJRE|MPJPE|MPJVE|Hand PE|Upper PE|Lower PE|Root PE|Jitter|
> |-|-|-|-|-|-|-|-|-|
> |SAGE|2.53|3.28|20.62|1.18|1.39|6.01|2.95|6.55|
> |SAGE(TMT)|2.47|3.14|20.01|0.95|1.30|5.79|2.87|7.44|
> |**MotionMAR**|2.39|2.82|16.23|0.83|1.22|5.13|2.58|5.17|
>
> >- (W4) ... details about VQVAE codebook quantization process ...
>
> Algorithm 1 (Encoding) and Algorithm 2 (Decoding) in Appendix B.1 describe the codebook quantization process. We summarize the process of TMT as follows.
> 1. Given an input motion latent feature of length $T$, we iteratively quantize it across $K=3$ temporal scales ($t_k \in \{T/4, T/2, T\}$) using a single, shared codebook.
> 2. At each scale $k$, the residual feature (initialized as the input latent) undergoes four steps: First, the residual feature is downsampled to the resolution $t_k$ via 1D linear interpolation. Next, the downsampled feature and the shared codebook embeddings are $L_2$\-normalized. The nearest codebook vectors for the downsampled feature are retrieved via cosine similarity. Then, the codebook vectors (the quantized vectors) are interpolated to the original length $T$. A 1D residual convolution block ($\Phi$) is applied to smooth temporal discontinuities. Finally, this smoothed quantized feature is added to the final reconstruction and subtracted from the current residual, yielding the input for the next, finer scale.
>
> >- (W5) ... why the authors have employed VQVAE ... vs that of a continuous regression paradigm.
>
> 1. The human motion generation community[1-3] recently adopted VQVAE due to its performance.
> 2. The AvatarPoser and AvatarJLM in Table 1 of our paper are based on **continuous regression models**. We find that methods (MotionMAR, SAGE) based on **discrete representations** significantly outperform these continuous methods in retaining realistic motion.
> 3. We design MotionMAR(Continuous), a MotionMAR variant that replaces the VQ-VAE with a continuous Autoencoder. In MotionMAR(Continuous), the autoregressive network regresses continuous latent vectors using MSE loss, rather than predicting discrete indices via Cross-Entropy. The experimental results show that modeling discrete tokens via VQ-VAE yields significantly better performance, achieving higher motion fidelity and lower reconstruction error than the continuous one.
>
> |Method|MPJRE|MPJPE|MPJVE|Hand PE|Upper PE|Lower PE|Root PE|Jitter|
> |-|-|-|-|-|-|-|-|-|
> |SAGE|2.53|3.28|20.62|1.18|1.39|6.01|2.95|6.55|
> |MotionMAR(Continuous)|2.53|3.37|26.01|1.62|1.49|6.10|2.97|11.67|
> |**MotionMAR**|2.39|2.82|16.23|0.83|1.22|5.13|2.58|5.17|
>
> [1] Stratified avatar generation from sparse observations. CVPR 2024
>
> [2] Tokenhmr: Advancing human mesh recovery with a tokenized pose representation. CVPR 2024
>
> [3] Momask: Generative masked modeling of 3d human motions. CVPR 2024

---

> > ### Author Rebuttal · Reviewer_bj8P · 2026-04-06
> >
> > I appreciate the authors' efforts in addressing my concerns, especially in regards to the additional results for SAGE+TMT and continuous vs discrete representations.
> >
> > My remaining concerns relate to clarity of presentation, which I recognize cannot be fully resolved during the rebuttal phase. Given the otherwise solid contributions, I will raise my score and recommend that the authors make a concerted effort to improve clarity in the final version.

---

> > > ### Author Response · Authors · 2026-04-08
> > >
> > > We sincerely thank the reviewer for evaluating our rebuttal and choosing to raise the score. We are glad the additional experiments, particularly the SAGE+TMT and continuous vs. discrete ablations, successfully addressed the reviewer's primary concerns.
> > >
> > > We agree with the reviewer's remaining notes on presentation and will implement the recommended formatting changes in the camera-ready version. To improve readability, we will move key citations to the Introduction and clearly list our core contributions. We will also update Figure 2 to the redesigned, explicitly labeled version provided in our response. To ensure the methodology is complete and easy to follow, we will summarize the progressive residual quantization steps of the TMT module directly in the main text and incorporate the new ablation results into the final manuscript.
> > >
> > > We appreciate the reviewer's constructive feedback, which has helped us significantly improve the paper.

---

### Decision · Program_Chairs · 2026-04-30

**Decision:**

Accept (regular)

**Comment:**

This paper proposes MotionMAR, a temporal coarse-to-fine autoregressive framework for reconstructing full-body human motion from sparse head-and-hand tracking signals. The method comprises three stages: a Temporal Multi-scale VQ-VAE that disentangles global trajectories from fine-grained dynamics, a Motion Autoregressive Network that performs next-scale prediction guided by a Scale-aware Control Module, and a Motion Refinement Network (BiGRU) that smooths quantization artifacts. Experiments on AMASS demonstrate state-of-the-art reconstruction accuracy.

It received mixed scores, i.e., 4, 3, 4, and 4. All four reviewers acknowledged the well-motivated core idea of adapting multi-scale autoregressive modeling from the vision domain to the temporal dimension of human motion. The main concerns across reviews fell into three categories: (1) presentation quality; (2) pipeline complexity, and (3) evaluation breadth. All these concerns have been resolved by a thorough rebuttal with extensive new experiments.

Overall, the paper makes a solid and timely contribution to sparse human motion reconstruction, backed by comprehensive experiments and a strong rebuttal.  The AC is happy to accept this paper. Congratulations! Please keep in the mind that the authors are strongly encouraged to incorporate all architectural details and hyper-parameters into the main text/supplementary, improve figure clarity, revise LLM-generated prose, and include the new ablation results (GORP evaluation, spatial-vs-temporal comparison, scale-count study) in the final camera-ready version.